# Recent Advances on Materials for Lithium-Ion Batteries

João C. Barbosa [1], Renato Gonçalves [2], Carlos M. Costa [1,*] and Senentxu Lanceros-Mendez [3,4,*]

1   Center of Physics, University of Minho, 4710-057 Braga, Portugal; joaocpbarbosa@live.com.pt
2   Center of Chemistry, University of Minho, 4710-057 Braga, Portugal; r.goncalves@quimica.uminho.pt
3   BCMaterials, Basque Center for Materials, Applications and Nanostructures, UPV/EHU Science Park, 48940 Leioa, Spain
4   IKERBASQUE, Basque Foundation for Science, 48009 Bilbao, Spain
*   Correspondence: cmscosta@fisica.uminho.pt (C.M.C.); senentxu.lanceros@bcmaterials.net (S.L.-M.)

**Abstract:** Environmental issues related to energy consumption are mainly associated with the strong dependence on fossil fuels. To solve these issues, renewable energy sources systems have been developed as well as advanced energy storage systems. Batteries are the main storage system related to mobility, and they are applied in devices such as laptops, cell phones, and electric vehicles. Lithium-ion batteries (LIBs) are the most used battery system based on their high specific capacity, long cycle life, and no memory effects. This rapidly evolving field urges for a systematic comparative compilation of the most recent developments on battery technology in order to keep up with the growing number of materials, strategies, and battery performance data, allowing the design of future developments in the field. Thus, this review focuses on the different materials recently developed for the different battery components—anode, cathode, and separator/electrolyte—in order to further improve LIB systems. Moreover, solid polymer electrolytes (SPE) for LIBs are also highlighted. Together with the study of new advanced materials, materials modification by doping or synthesis, the combination of different materials, fillers addition, size manipulation, or the use of high ionic conductor materials are also presented as effective methods to enhance the electrochemical properties of LIBs. Finally, it is also shown that the development of advanced materials is not only focused on improving efficiency but also on the application of more environmentally friendly materials.

**Keywords:** electrodes; solid polymer electrolytes; separators; battery systems





## 1. Introduction

The search for more efficient and sustainable energy storage devices is a growing need and a fruitful research field, based on the increasing mobility of society. Industrial production and mobility require significant quantities of energy, mostly relying on fossil fuels, as coal or petroleum, and more recently natural gas and nuclear fission [1]. Nowadays, with the growing awareness with respect to environmental issues, renewable energy sources gained significant interest as they allow to obtain green energy, with the reliance in inexhaustible sources, such as wind, water or sun [2,3]. However, the use of renewable energies is limited by their irregularity, which does not warrant a constant energy supply [4]. In this context, the integration of renewable energies with efficient energy storage systems (ESS) arises as a potential solution regarding this issue.

These ESS can vary depending on the application needs, batteries being the most used systems worldwide. Since the voltaic pile, created in 1800 [5], until the commercialization of the first lithium-ion battery (LIB) in 1991, battery technology has undergone strong developments, with significant improvements on capacity, durability, and reversibility (Figure 1).

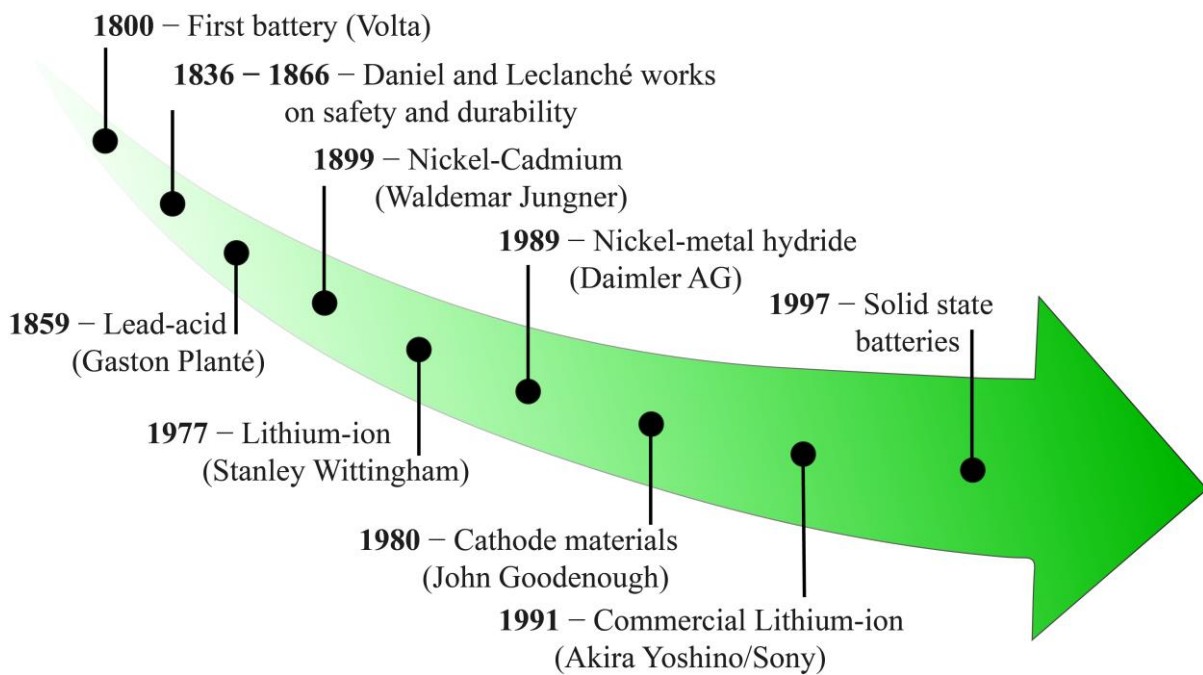

**Figure 1.** Advances in battery system development over the years.

LIBs are the most commonly used ESS in modern society, mainly due to their high specific capacity, making them appropriate for small and light portable devices without limiting their performance. LIBs are also characterized by prolonged cycle life and no memory effects [6]. These are important advantages that increased the use of LIBs, leading to a progressive replacement of previous technologies, such as nickel–cadmium and nickel–metal hydride batteries, which are less efficient, in particular for small device applications. In fact, the developments in the area of ESS further supported the strong evolution of the market of portable equipment with increased and improved features, capacity and smaller sizes. LIBs are present in a large variety of commonly used devices including smartphones, tablets, laptops, grid stabilization, electric vehicles, and a vast range of other electronic systems [7].

In this constant growing field, it is extremely important to keep up with the high number of works that are published every year. Thus, a frequent systematization and review of the literature is needed in order to comparatively analyze newly developed materials, integration strategies, and battery performance data to properly design further developments in the field. The main focus of the research community in the field nowadays is the search for new high-performance materials, but also to address environmental questions, by finding more sustainable and green solutions. In the present work, the LIB constitution and working principles are described, presenting the most recent solutions for each component of the battery separately. In particular, the latest developed materials and production/modification strategies are considered.

## 2. Lithium-Ion Batteries

As stated before, LIBs are the most used energy storage devices worldwide, due to their versatility and high capacity, when compared with other ESS [8]. These devices convert chemical energy into electrical energy and vice versa, with high energy density, making them particularly suitable for smaller devices [9]. However, some issues related to the costs, progressive degradation of the components, and the necessity to maintain the voltage and current within safe limits need to be addressed for the LIB technology to be fully optimized [10,11].

The lithium-ion technology is somewhat recent, as it only appeared in the second half of the twentieth century. Stanley Whittingham in 1977 was responsible for the development

of the first device [12]. The safety problems associated with lithium dendrites growth were solved in the 1980s by John Goodenough with the development of new cathode active materials and carbonaceous anode materials, avoiding the use of lithium metal anodes [13–15]. The first commercial battery was developed by Sony, using the work of Akira Yoshino [16], and it was placed in the market in 1991. These three researchers won the Nobel Prize in Chemistry in 2019 due to the relevance of their work in the essential area of energy storage. The research work in the battery field has been always accompanied by efforts on developing solid-state technology [17], in order to eliminate the liquid components of the battery structure, leading to safer devices, despite the difficulty of obtaining consistent results.

The typical constitution of a battery includes two electrodes (cathode and anode) and a separator membrane soaked in an electrolyte solution [18]. Each electrode is composed by an active material, a conductive material, and a binder. Both electrodes share similar binders and conductive materials, with the function of warranting the structural cohesion and the adhesion to the structure, in the case of the binder, and increasing the electric conductivity of the electrode, in the case of the conductive material. The battery capacity is determined by the active material, with the ability to deliver or store lithium-ions, in the cathode and the anode, respectively [19]. The current collectors are made of different metals and are placed in each electrode, contributing to the electronic conductivity and potential stability of the cell. The potential difference generated between the electrodes allows for the redox reactions to occur, which represents the basic working principle of the batteries [20]. Between the electrodes, a polymeric porous membrane, the separator, is placed to avoid the physical contact, preventing short circuits. The separator is typically soaked in an electrolyte solution, which increases the ionic conductivity of the system, allowing for an easier $Li^+$ diffusion [21]. More recently, this design of the separator/electrolyte system is being replaced by a solid electrolyte, which combines the physical barrier function of the separator and the high ionic conductivity of the electrolyte, allowing the elimination of the liquid components from the battery structure [22]. However, there are still major drawbacks in this approach that limit battery performance, such as low ionic conductivity and difficult interfacial compatibility with the electrodes [23].

The function of a LIB is based in redox reactions. During the charging process, energy is provided to the system accompanied by a flow of electrons from the cathode to the anode, making the $Li^+$ ions migrate from the cathode to the anode in order to compensate the created charge difference. This reaction is reverted during the discharge process. In this case, the $Li^+$ ions return to their original state, with the corresponding release of energy, which can be used for different applications. This flow of $Li^+$ and electrons is described by Equation (1):

$$LiAM \leftrightarrow AM + xLi^+ + xe^- \tag{1}$$

where AM is the active material and $e^-$ is an electron.

The most common LIBs use different metal oxides, such as $LiFePO_4$ [24], $LiCoO_2$ [25], or $LiMnO_2$ [26], as active materials for the cathodes. The selection of the active material is dependent on the specific application, as each one allows for different operational voltage [27]. For the anode, the most common active materials are carbon-based materials, such as graphite and silicon-based materials [28]. At the separator level, different kinds of polymers such as poly(ethylene) (PE) [29], poly(propylene) (PP) [30], or poly(vinylidene fluoride) (PVDF) [31] are successfully applied due to their properties, including high mechanical and thermal resistance, suitable porosity, electrolyte wettability, and electrochemical stability [32]. The electrolyte solution is usually composed by a lithium salt, lithium hexafluorophosphate ($LiPF_6$), dissolved in organic carbonates, such as ethylene carbonate (EC) and dimethyl carbonate (DMC) or diethyl carbonate (DEC) [33]. In the field of solid electrolytes, they can be inorganic or organic. The inorganic ones are composed by ceramic crystalline materials as LISICON [34], NASICON [35], or perovskites [36], and they usually possess high ionic conductivity but have significant limitations when it comes to interfacial compatibility [23]. Solid polymer electrolytes comprise a polymer matrix and

one or more fillers incorporated in their structure [37]. They show high mechanical and thermal stabilities but low ionic conductivity, being difficult to find the balance between the type and amount of fillers to optimize the abovementioned properties [38].

Other technologies based on lithium include lithium-air and lithium-sulfur batteries. These batteries represent promising options due to their higher theoretical capacities when compared to conventional LIBs. However, limitations such as the difficulty to obtain perfectly reversible reactions, control of the volume changes, and to warrant suitable ionic conductivity are holding back the full potential of these kind of batteries [39,40].

Beyond lithium, other promising battery technologies are being intensively studied and developed, as presented in Table 1.

**Table 1.** Comparative analysis between different battery technologies. The variability of data for the same type of battery is due to the different active materials used for a given technology.

| Technology | Specific Energy (Wh/kg) | Number of Cycles | Efficiency (%) | Voltage (V) | Ref. |
|---|---|---|---|---|---|
| Lithium-sulfur | 500 | ~500 | 85 | 3 | [40] |
| Lithium-ion | 100–265 | 1000–2000 | 99.9 | 3.6 | [41] |
| Lithium-air | 3860 | 700 | 65 | 2.91 | [42] |
| Potassium-ion | 120–170 | ~4000 | >90 | 2.0–4.3 | [43] |
| Magnesium-ion | 77 | ~2000 | ~95 | 1.1 | [44] |
| Sodium-ion | 85–125 | ~500 | >90 | 2.7–3.2 | [45] |

These technologies aim to overcome some of the problems associated with lithium, such as its scarcity [46]. Thus, the abundance of sodium and magnesium could represent an effective and cheaper alternative if the problems associated with their high reactivity could be overcome [47]. Potassium has the advantage of high voltage operation; however, its low melting point can be an issue in the case of higher operation temperatures [48]. Beyond batteries, there are other ESS, such as fuel cells or hydrogen-based technologies, that can be successfully integrated with the most common batteries. However, there is a long path to go in research and development for these technologies to achieve their full potential [49]. This means that despite the alternatives to and the limitations of the commonly used LIBs, these devices are still the most appropriate, at least until further developments are made in other technologies. With this purpose, the research and development of advanced, more efficient, safer and environmentally friendlier materials for energy storage devices is a very relevant area nowadays.

In the next sections, the latest advances in the materials used for the different battery components will be provided, with particular focus in the materials that offer potential solutions for the previously mentioned LIBs issues.

## 3. Materials for Electrodes

Regardless of the type of electrode, its basic constituents are the active material, the conductive material, and the polymer binder. The microstructural characteristics of lithium-ion battery electrodes also determine their performance [50].

In the electrode composition, the amount of active material typically varies between 60 wt % and 95 wt %, the conductive material varies between 3 wt % and 30 wt %, and the polymer binder varies between 2 wt % and 25 wt %, where the proportion of 80 wt %/10 wt %/10 wt % for active material, conductive material, and polymer binder is the most used by the scientific community [51]. Basically, the difference between the two electrodes (anode and cathode) is the active materials; therefore, this review focuses on the recent advances for each of the active materials divided by the type of electrode.

Several issues can affect the performance and durability of the electrode materials, including volume expansion of the electrode during the charge–discharge process [52,53], mechanical failure of the electrodes due to external mechanical/thermal loadings [54,55],

thermal failure caused by the battery overheating [56], and tortuosity/percolation limitations [57]. Some suggested solutions for these issues are presented in the following.

### 3.1. Active Anode Materials

Anode active materials can be structures of different types, their main characteristic being the capacity, electrical conductivity, mechanical stress, and structural stability, which plays a relevant role in the durability of the material through the Li$^+$ intercalation–de-intercalation reactions [58].

In addition to the gravimetric and volumetric capacities of each anode active material, other parameters, such as the average voltage range, porosity, and irreversible capacity loss are relevant in determining the performance of the anode. Typically, the main active materials for the anode are carbonaceous materials, alloys based on Si, Sn, Al, Ga, Ge, Pb, and Sb, metal oxides, and metal chalcogenides, among others. Figure 2 shows the voltage range vs. Li/Li$^+$ as a function of charge capacity of these anode active materials [59,60].

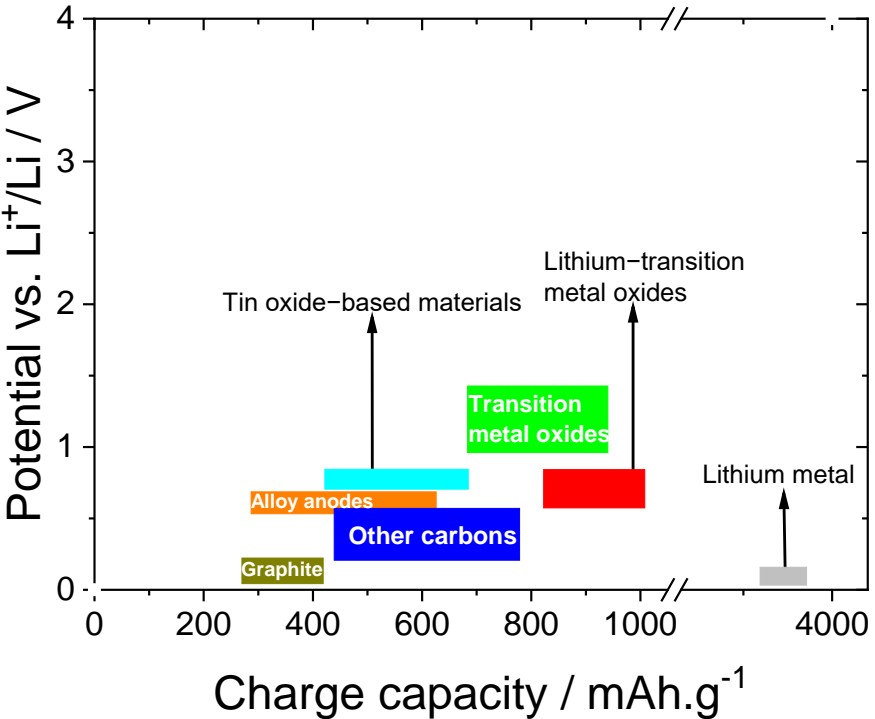

**Figure 2.** Voltage range vs. Li/Li$^+$ as a function of the charge capacity of different anode active materials (adapted from [61]) [59].

Carbonaceous materials are the most widely used active materials for the anode, including graphite and carbon nanotubes, among others, mainly due to the reversibility, low cost, low volume variation during the intercalation/deintercalation process, and high number of charge/discharge cycles. One of the main disadvantages of graphite is the irreversible loss of capacity during the first charge–discharge cycle to form a stable interface between the electrolyte and the graphite, called the solid electrolyte interface (SEI). Despite the fact that the SEI prevents the degradation of the battery materials, it also increases the overall resistance of the system, causing a loss of performance [62,63].

Conversion-type transition-metal compounds (MaXb, M = Mn, Fe, Co, Ni, Cu, X = O, S, Se, F, N, P, etc.) are attractive active materials for anodes due to their high theoretical capacity, tunable operation voltages, and the diversity of chemical composition and phases, allowing tuning materials characteristics. The main disadvantage is their poor intrinsic conductivity [64].

Recently, advances on anode active materials are focused on their compositional variation and design, taking into account that the low-dimensional, inter-spatial, and composite

design of materials such as ordered-array, cross-aligned, or alternating-layer structure affect the lithium-ion and electrons transport within the electrodes, solid electrolyte interphase, and reversibility, among others [65]. In the following, advances in this area are divided by the composition of each material.

### 3.1.1. Carbon and Metal Alloys-Based Anode Materials

Graphite is widely used in the anode electrode, although, to improve even more its energy density, graphite intercalation compounds (GICs) have been developed allowing enhancing the electrochemical performances. Advanced anode materials based on cobalt chloride–ferric chloride–graphite bi-intercalation compounds ($CoCl_2$-$FeCl_3$-GICs) have been synthetized through the molten-salt method, showing a high capacity of 1033 mAh·$g^{-1}$, with a retention rate of 94.2% at 200 mA·$g^{-1}$ and, after 350 cycles under 1000 mA·$g^{-1}$, the charge capacity reached 536 mAh·$g^{-1}$, as shown in Figure 3a [66].

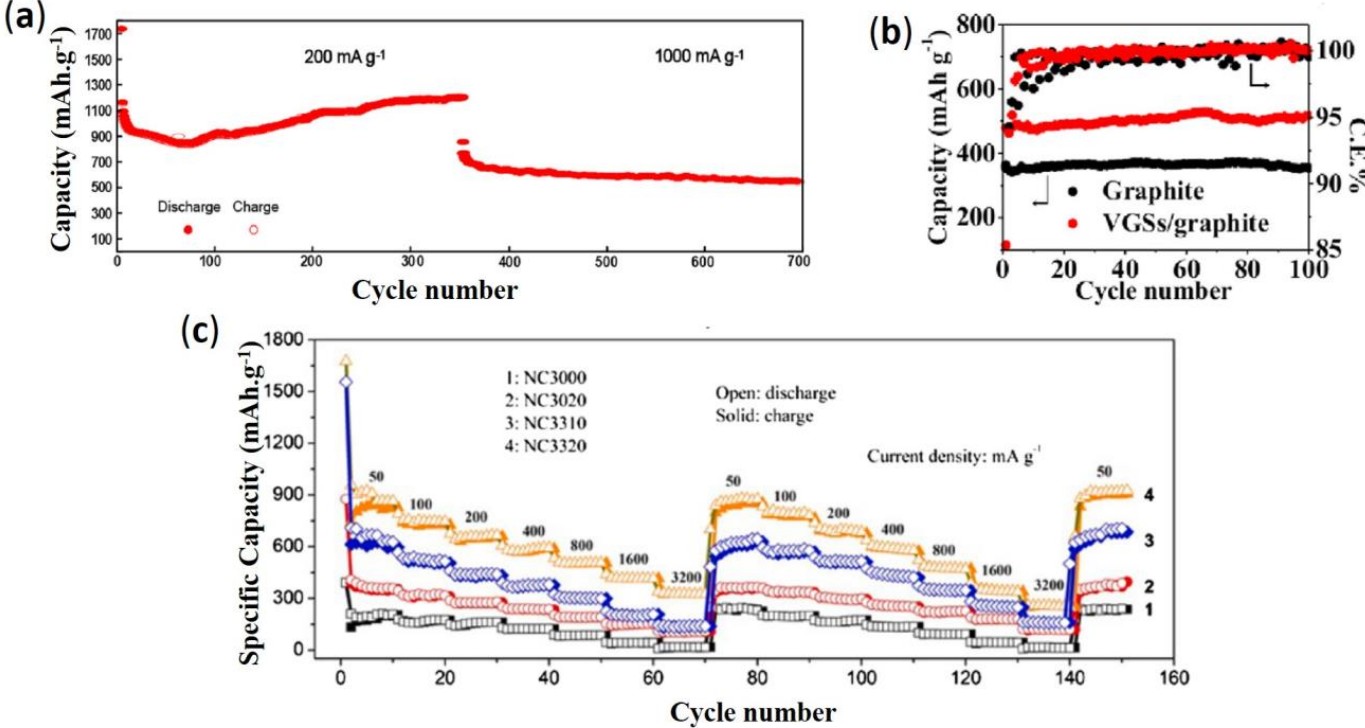

**Figure 3.** (**a**) Long-term cycling performance of $CoCl_2$-$FeCl_3$-GICs at 200 and 1000 mA·$g^{-1}$ [66]. (**b**) Cycling curves of graphite and vertical graphene sheets (VGSs)/graphite at 0.2 C [67]. (**c**) Rate performance at various current densities for the different nitrogen-doped graphene-like carbon nanosheets samples [68].

In order to improve the intrinsic electrical conductivity of graphite, a new material based on VGS/graphite was produced with high specific capacity. The electrochemical performance is shown in Figure 3b [67]. Using coffee grounds as a carbon and nitrogen source, and $CaCO_3$, $Fe(NO_3)_3$, and their derivatives as structural templates and graphitization catalysts, nitrogen-doped graphene-like carbon nanosheets were prepared in the form of 3D porous architecture. The corresponding battery performance is shown in Figure 3c [68]. Alloys are also used as anode active materials, since they can adapt to volume changes during lithiation. Thus, the ternary Li-Sb-Sn system has been developed being able to uptake up to 15 at % Li without changing the crystal structure [69].

### 3.1.2. Silicon-Based Anode Materials

Silicon-based materials are characterized by a high specific capacity, over 10 times when compared with the one of graphite [70]. On the other hand, they exhibit large volume changes and form an unstable SEI. Recent advances in these materials are focused on

solving those issues by developing advanced silicon-based materials [71]. The combination of silicon nanostructures with carbon structures is the main alternative to circumvent the disadvantages of silicon-based anodes [72] such as a coral-like porous Si/C (CLP-Si/C), which exhibit a stable capacity of 990.6 mAh·g$^{-1}$ and can be kept after 100 cycles at a rate of 250 mA·g$^{-1}$ [73]. Other interesting material is amorphous carbon cascade-coated nano-silicon, with 89% initial coulombic efficiency (ICE) and high reversible capacity of 874.5 mAh·g$^{-1}$ after 300 cycles. The processing of this material is shown in Figure 4a [74]. Free-standing N-doped porous carbon nanofibers sheathed pumpkin-like Si/C composites (Si/C-ZIF-8/CNFs) [75], Si/carbon nanotube microspheres (Si/CNTsS) prepared by chemical vapor deposition [76], silicon (Si)/carbon(C) composites prepared by mixing appropriate concentrations of hydrocarbon resin, Si powder, and polyacrylic acid [77], and Si nanoparticles confined within a conductive 2D porous Cu-based metal–organic framework (Cu$_3$(HITP)$_2$) [78] represent interesting examples for the advanced materials developed in this area. Particularly interesting are Si@Cu$_3$Si nanocomposites, which show a delivering charge/discharge capacity of 1000 mAh·g$^{-1}$ even after 300 cycles [79]. Si powders with electrical conductivity of 1.04 µs·cm$^{-1}$ have been obtained from p-type solar grade broken Si wafers with electrical resistivity of about 1–10 Ω·cm via ball-milling and tested for anode electrodes. The charge-specific capacity of p-doped Si holds 1920.3 mAh·g$^{-1}$ after 50 cycles at 0.84 A·g$^{-1}$, when the charge current density increased to 21 A·g$^{-1}$ (only taking 12 min for charging), and it is still maintains at 1758.5 mAh·g$^{-1}$ after 50 cycles [80].

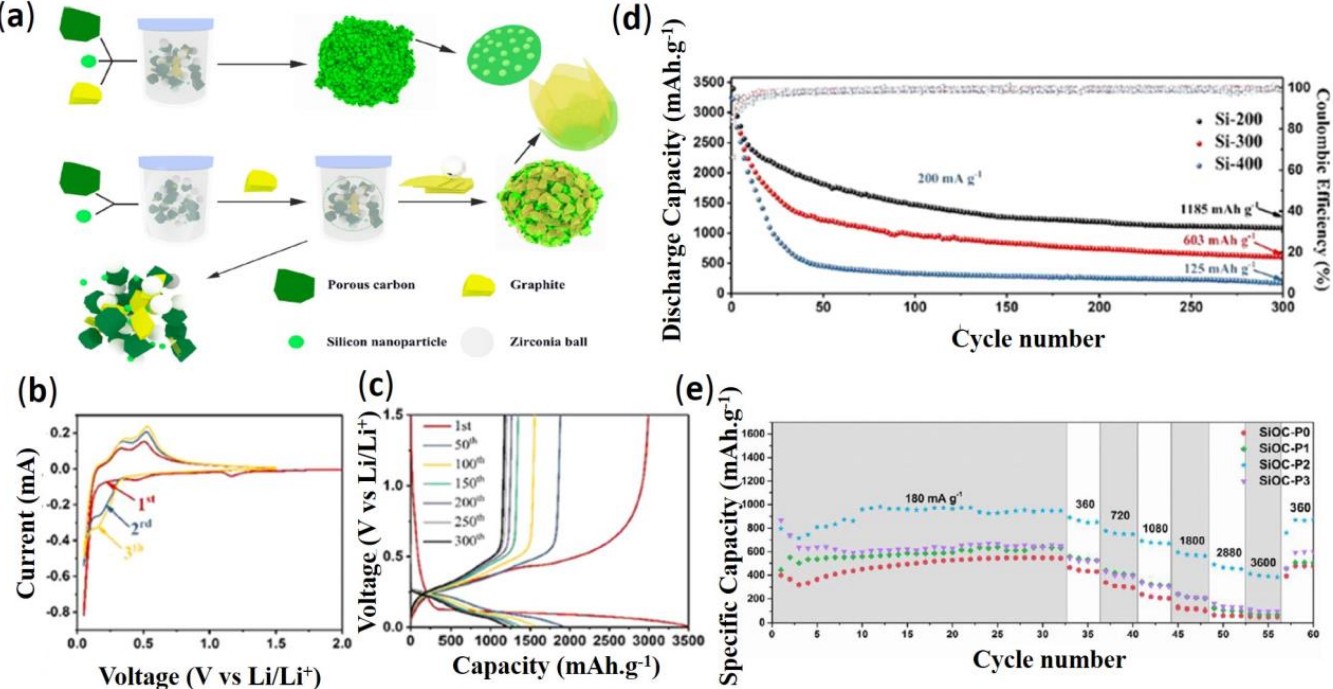

**Figure 4.** (**a**) Schematic illustration of the preparation process of amorphous carbon-coated silicon [74], (**b**) CV curves of Si-200 at a scan rate of 0.1 mV·s$^{-1}$; (**c**) Galvanostatic discharge/charge profiles of Si-200 at 0.2 A·g$^{-1}$; (**d**) Cycling performance of various nano-Si at 0.2 A·g$^{-1}$ [81]; and (**e**) Rate capability of the different silicon oxycarbide (SiOC) samples measured at various C-rates from 180 to 3600 mA·g$^{-1}$ [82].

A MgH$_2$-AlCl$_3$-SiO$_2$ melt system was developed to synthesize nano-Si through the reduction of SiO$_2$ by MgH$_2$ into molten AlCl$_3$. The obtained nano-Si product shows an average particle size of 22.4 nm and exhibits a superior electrochemical storage capacity of 1185 mAh·g$^{-1}$ over 300 cycles at 0.2 A·g$^{-1}$ and a low thickness variation of 14.5% at 2 A·g$^{-1}$ over 500 cycles, as shown in Figure 4b–d [81].

Silicon oxycarbide (SiOC, SiO$_n$ C$_{4-n}$ ($0 \leq n \leq 4$)) has been developed for anode material through the introduction of PSS-Octakis (dimethylsilyloxy) silsesquioxane (POSS)

into the synthesis process of SiOC. The rate performance of these materials is shown in Figure 4e [82]. Silicon oxycarbide (SiOC) was also synthetized by pyrolysis using silicone oil and phenyl group-containing additives (divinylbenzene -DVB) as precursor, leading to a high reversible capacity (550 mAh·g$^{-1}$ at 200 mA·g$^{-1}$) [83].

### 3.1.3. Conversion-Type Transition-Metals and Their Composites-Based Anode Active Materials

Transition metals are another excellent option for the development of anode electrodes for LIBs due to their high theoretical capacities, low cost, and easy availability. The focus on these materials is the development of new compositions and the improvement of their electrical conductivity.

Various transition metals such as cerium vanadate ($CeVO_4$) [84], $Co_{0.85}Se$ [85], $CuSi_2P_3$ [86], iron oxide ($Fe_2O_3$) [87], $Mn_3O_4$ [88], octahedral nanostructured $Cu_2WS_4$ [89], copper oxide ($Cu_2O$) [90], $Li_3VO_4$ particles [91], NiO nanocrystals [92], nano-$Mn_2O_3$ particles [93], $MnV_2O_4$ particles [94], $SnO_2$ [95], molybdenum sulfide ($MoS_2$) [96], $MoSe_2$ [97], Sn-based material ($SnFe_2O_4$) [98], $SnS_2$ nanoflowers [99], $SiO_2$ [100], $SnO_2$ [101], $SnO_2@ZrO_2$ [102], and $WS_2$ heterostructures [103] have been developed and doped with carbonaceous materials (carbon agents, carbon nanotubes, and graphene oxide, among other). This carbonaceous doping has the objective of increasing the electrical conductivity and, consequently, the electrochemical properties through conductive pathways to facilitate charge transport and structural buffer space to accommodate volume variations. Other active materials with interesting properties are Columbite $CuNb_2O_6$ with 154.9 mAh·g$^{-1}$ at the ultra-large current rate of 5 A·g$^{-1}$ [104], $TiNb_2O_7$ with specific capacity of 220 mAh·g$^{-1}$ after 500 cycling at 0.5 C [105], and ammonium manganese phosphate hydrate (NMP) [106].

A widely used transition metal oxide is iron oxide ($Fe_2O_3$), considering the low cost and abundant reserves. In order to improve its performance, several approaches have been taken, including in situ encapsulation of $\alpha$-$Fe_2O_3$ nanoparticles into micro-sized $ZnFe_2O_4$ capsules [107], a new synthesis method based on chemical precipitation with the sulfuric acid leaching liquor of tin ore tailings as an Fe source [108] and a new material based on Rosa roxburghii-like hierarchical hollow sandwich-structure C@$Fe_2O_3$@C microspheres [109]. In addition, $Co_3O_4$/Co in situ nanocomposites were synthetized to improve SEI [110]. $CoS_2$-MnS@CNT have been produced with excellent rate performance (1620 mAh·g$^{-1}$ at 100 mA·g$^{-1}$) and high reversible capacity (1327 and 927 mAh·g$^{-1}$ at 100 and 1000 mA·g$^{-1}$, respectively, after 100 cycles) [111]. In addition to carbon, nickel has been used for doping $Co_9S_8$@ZnS composites in order to improve the overall battery performance [112]. A new material based on hollow core–shell structured CNT/PAN@$Co_9S_8$@C coaxial nanocables have been produced for anode active material with the aim of providing more channels for Li$^+$ ions/electrons diffusion and relieving volume swelling during the charge/discharge process. This material exhibits excellent cycling performance (>700 mAh·g$^{-1}$ at 0.1 A·g$^{-1}$) [113].

The novel mixed Co/Mn vanadates have been obtained by balancing performance and cost through the doping of Mn for cobalt vanadate through an easy hydrothermal reaction where the optimized element ratio is found for 67% Mn-based vanadates. This material shows a high initial discharge capacity of 1193 mAh·g$^{-1}$ and maintains 935 mAh·g$^{-1}$ after 500 cycles at 0.5 A·g$^{-1}$ [114].

A multifunctional heterostructure, $MoS_3$-$Ti_3C2T_x$, comprising a functionalized MXene ($Ti_3C_2T_x$) and amorphous $MoS_3$ has been prepared by a scalable electrostatic self-assembly method, $MoS_3$-$Ti_3C_2T_x$ offering an excellent reversible capacity of 1043 mAh·g$^{-1}$ at 200 mA·g$^{-1}$ and exhibiting a capacity of 568 mAh·g$^{-1}$ at a current density of 2 A·g$^{-1}$ after 1000 cycles. Figure 5a shows the cycling performance of this material [115].

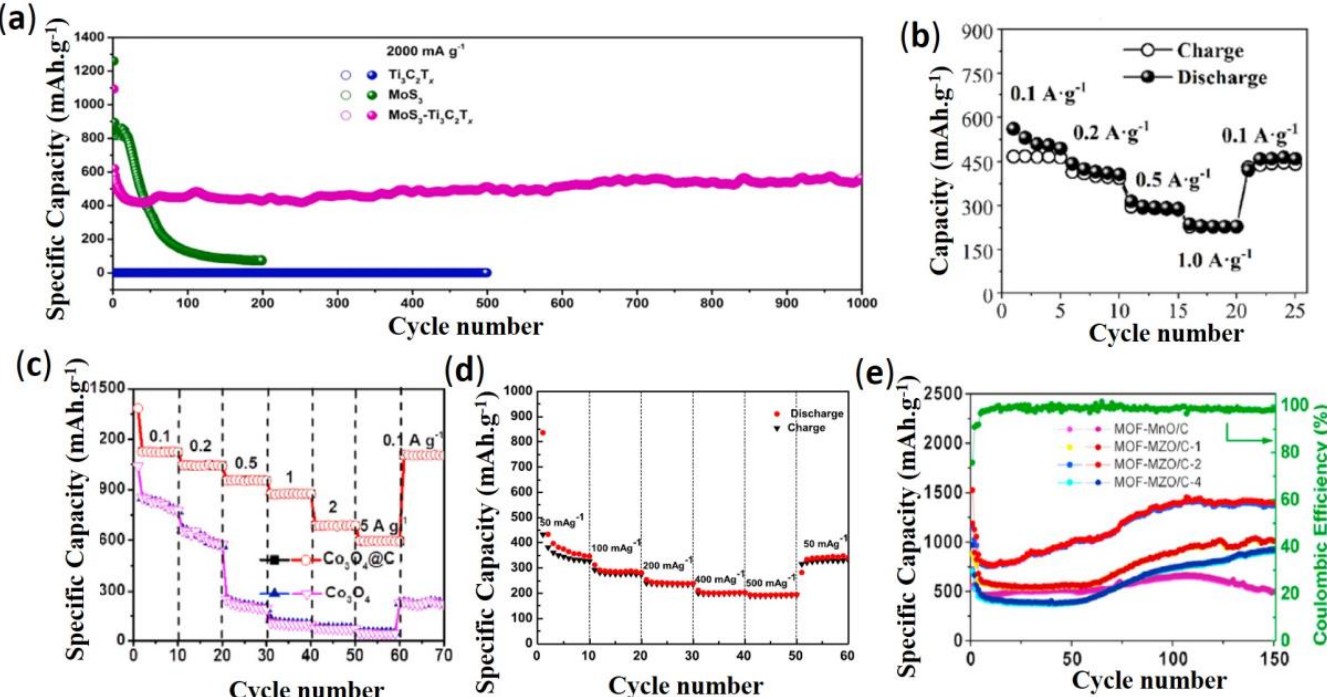

**Figure 5.** (**a**) Cycling performance of MoS$_3$-Ti$_3$C$_2$T$_x$ at 2000 mA·g$^{-1}$ [115], (**b**) Rate capability of CoO/Co$_{1.94}$P@CP [116], (**c**) Rate performance of Co$_3$O$_4$@C and bare Co$_3$O$_4$ at various current densities from 0.1 to 5 A·g$^{-1}$ in the voltage range of 0.01–3.0 V [117], (**d**) Rate performance of TiO$_2$/C composite-fibers electrode at various current densities from 50 mA·g$^{-1}$ to 500 mA·g$^{-1}$ over 10 cycles [118], and (**e**) Cycling tests of MOF-MnO/C and MOF-MZO/C electrodes at 0.2 A·g$^{-1}$ [119].

A hetero-structured few-layer MoS$_2$-coated MoO$_2$ (MoS$_2$@MoO$_2$) has been fabricated for anode applications. This material shows a high reversible specific capacity of 1263 mAh·g$^{-1}$ after 40 cycles at 0.1 A·g$^{-1}$ [120]. In addition, nanocomposites based on Fe$_2$O$_3$–TeO$_2$–MoO$_3$ allow improving the electronic conductivity and, consequently, lead to excellent battery performance [121]. SnO$_2$/NiFe$_2$O$_4$/graphene (SNG) nanocomposites also lead to a high discharge capacity of 613 mAh·g$^{-1}$ at 800 mA·g$^{-1}$ after 100 cycles [122].

Another suitable nanocomposite is CoP-Co$_2$P/Ti$_3$C$_2$ based on MXene, which effectively improves the cycle stability performance compared to Ti$_3$C$_2$ [123]. A new material based on MXene results from the intersection of TiO$_2$ nanosheets using a molten-salt method, followed by the generation of dispersed Li$_3$Ti$_2$(PO$_4$)$_3$ NCs. This composite allows improvement of structural stability while showing a high discharge capacity of 204 mAh·g$^{-1}$ at 50 mA·g$^{-1}$ [124]. In addition, a new anode material based on CoO/Co$_{1.94}$P nanocrystals wrapped within carbon polyhedron (CoO/Co$_{1.94}$P@CP) heterostructure has been prepared, and its cycling performance is shown in Figure 5b [116].

In order to improve the cycling performance of niobium oxide (Nb$_2$O$_5$), a new material based on aspergillus oryzae spore carbon (ASC) with niobium oxide has been developed to form ASC/Nb$_2$O$_5$ composites, The material is synthetized by the solvothermal (ST) method and offers discharge capacities of 189 and 65 mAh·g$^{-1}$ as the current density increases from 0.2 to 20 C [125].

Another transition metal of mixed valence spinel cobalt oxide (Co$_3$O$_4$) has been used for anode electrodes, but it has low electronic conductivity and large volume expansion. To solve these disadvantages, carbon-coated porous Co$_3$O$_4$ polyhedrons with (220) facets are produced through a hydrothermal method. The discharge capacity of this material reaches 1463 and 596 mAh·g$^{-1}$ at 100 and 5000 mA·g$^{-1}$, respectively, as shown in Figure 5c [117].

Furthermore, TiO$_2$ particles have been used to improve the electrical conductivity of the anode. TiO$_2$/carbon composite-fiber anodes are developed through centrifugal spinning of TiS$_2$/polyacrylonitrile (PAN) precursor fibers and subsequent thermal treatment. This material allows increasing the specific capacity, improves stability, and enhances

electrochemical performance compared to $TiO_2$. The corresponding rate performance is shown in Figure 5d [118]. Another material based on $TiO_2$ is constituted by multi-role $TiO_2$ coated on carbon@few-layered $MoS_2$ (CMT) nanotubes, showing excellent long-term cycling performance [126].

Considering their properties, metal–organic framework (MOF) derivatives are increasing their applicability for anode materials, and MnO/ZnO@C nanohybrids have been developed with superior lithium storage capabilities, reaching a reversible capacity of 1396 mAh·$g^{-1}$ at 0.2 A·$g^{-1}$ with an initial coulombic efficiency higher than 75%, as shown in Figure 5e [119]. In addition, hollow urchins Co-MOF with fluorine (F) doping on reduced graphene oxide (rGO) was synthetized using a solvothermal reaction with excellent reversible capacity (1202.0 mAh·$g^{-1}$ at 0.1 A·$g^{-1}$) [127]. Finally, also related to MOFs, a new $Co_3O_4$/Co@N-C composite was developed for anode applications. This structure allows improving the electrical conductivity and acts as a buffer medium to alleviate the volume change [128]. Furthermore, it allows the development of new polyoxometalate-based metal–organic frameworks (POMOFs) of NAU3–6 with various architectural features [129].

### 3.2. Active Cathode Materials

For cathode electrodes, the main characteristics of the active materials are high reactivity with lithium, high voltage, easy intercalation and desintercalation of $Li^+$ ions during the charge and discharge process, i.e., lithium diffusion channels and low volume change, being good electronic conductors, stable in contact with the electrolyte solution, and having low cost. The most commonly used active materials for the cathode are lithium cobalt oxide ($LiCoO_2$, LCO), lithium manganese oxides ($LiMnO_2$ and $LiMn_2O_4$, LMO), lithium iron phosphate ($LiFePO_4$, LFP), lithium nickel cobalt oxide ($LiNi_{1-x}Co_xO_2$ ($0.2 \leq x \leq 0.5$), LNCO), lithium nickel manganese cobalt oxide ($LiNi_{1/3}Co_{1/3}Mn_{1/3}O_2$, LNCMO), and lithium nickel manganese oxide ($LiNi_{0.5}Mn_{0.5}O_2$, LNMO). Table 2 shows the crystal system, specific capacity, and voltage range for each active material.

**Table 2.** Most commonly active cathode active materials used for lithium-ion battery applications [130].

| Cathode Active Material | Crystal System/Space Group [Point Group] | Specific Capacity/mAh·$g^{-1}$ | Typical Voltage Range/V |
|---|---|---|---|
| $LiCoO_2$ | Orthorhombic/R3m [$C_{3V}$] | 274 | 2.5–4.45 |
| $LiFePO_4$ | Orthorhombic/Pnma [$D_{2h}$] | 170 | 2.5–4.2 |
| $LiMn_2O_4$ | Cubic/Fd$\bar{3}$m [$O_h$] | 148 | 3.0–4.3 |
| $LiNiO_2$ | Trigonal/R3m [$C_{3V}$] | 275 | 3.0–4.3 |
| $LiNi_{1-x}Co_xO_2$ ($0.2 \leq x \leq 0.5$) | Rhombohedral/R3m [$C_{3V}$] | ~275 | 3.5–4.3 |
| $LiNi_{1/3}Mn_{1/3}Co_{1/3}O_2$ | Rhombohedral/R3m [$C_{3V}$] | 278 | 2.3–4.3 |
| $LiNi_{0.5}Mn_{1.5}O_2$ | Trigonal/R3m [$C_{3V}$] | 147 | 3.5–4.9 |

Recent advances in cathode active materials are focusing on the optimization of particle size, morphology, specific functionalization, doping with different elements, and developing composites with different particles and coating. All these optimization strategies are focusing on the improvement of electronic and thermal properties, to stabilize the particle with respect to the electrolyte, to optimize synthesize methods (as sol–gel synthesis and co-precipitation), and to improve the mechanochemical activation [130,131].

$LiCoO_2$ is widely used in portable applications, such as smartphones, watches, or computers. In fact, it was applied in the first battery in 1991 by Sony due to its high volumetric energy density and reliability. The main disadvantage of this material is associated to the fact that its structure is not stable at high voltage (>4.2 V vs. Li/$Li^+$). Improvements are relying on doping with different elements, surface coating with $Li_3NbO_4$ and $Co_3O_4$ layers [132], and also through electrolyte optimization [25]. Doping with transition metal ions in $LiCoO_2$-based batteries has been shown to improve electrochemical properties due

to the distorted local structure promoted by the impurity ions [133]. $LiFePO_4$ (LFP) is an active material of particular interest to be applied in electric vehicles. To improve its performance, a carbon coating has been placed using biomass of phytic acid (PhyA) as a novel phosphorus source to replace traditional phosphoric acid [134], $LiFePO_4$/multi-walled carbon nanotube (MWCNT) composites have been developed by a hydrothermal process [135], zinc oxide and carbon co-modified $LiFePO_4$ nanomaterials (LFP/C-ZnO) have been produced by a hydrothermal method [136], and $LiFePO_4$/C have been implemented as active cathode materials [137]. Moreover, for LFP, the volumetric capacity has been increased by hot isostatic pressing, isostatic pressure facilitating the electrolyte to penetrate into the voids among LFP particles, which improves $Li^+$ ion diffusion [138].

A novel synthesis of $LiFe_{0.25}Mn_{0.75}PO_4$/C@reduced graphene oxide (rGO) has been developed with the aim of reducing the charge-transfer impedance of the electrode and improving the conductivity and electrochemical properties. With a specific discharge capacity of 143.8 and 139.8 mAh.g$^{-1}$ at 1 C and 2 C, respectively, this material shows excellent electrochemical reversibility [139]. $LiFeBO_4$ (LFeB) has received special attention considering its high theoretical capacity of 220 mAh·g$^{-1}$. To further improve its performance, a new active material, $LiFeBO_3$- x$F_2$ x (LFeBF, x = 0.05, 0.1, 0.2, 0.3, and 0.5), has been synthetized by a solid-state reaction, the fluorine substitution at the oxygen site of LFeB leading to an improvement in discharge capacity [140].

$LiMn_2O_4$ (LMO) is another active material widely used in LIBs, and its poor cycling behavior has been improved by doping with Ni, Cu, and Co through sol–gel synthesis. The $LiMn_{1.5}Ni_{0.5}O_4$-based cathodes show a decrease in the total half-cell resistance after cycling and excellent electrochemical stability [141]. In addition, for this active material, nanocomposites with graphene oxide have been developed for enhanced electrochemical performance [142]. In order to improve the spinel deterioration, a rock salt type $Li_2Nb_{0.15}Mn_{0.85}O_3$ was synthetized, its structure showing no tendency for spinel deterioration or cation ordering, even with massive lithium vacancies [143]. Furthermore, the spinel $LiMn_{1.5}Ni_{0.5}O_4$ has been coated with aluminum oxide to strongly influence the charge–discharge performance [144] and has been doped with vanadium to improve its thermal stability and cyclability [145]. $LiNi_{0.4}Mn_{1.6}O_4$ has been also doped with Ti, where the strong Ti-O bonds reinforce the oxygen lattice and stabilize the crystal structure during the electrochemical reactions process [146].

$LiNi_{0.8}Co_{0.1}Mn_{0.1}O_2$ (NCM811) is a promising active material suitable for electric vehicles based on its high specific energy and power density [147], although it shows the disadvantages of mixed cation discharge, poor thermal stability, and cycling performance [148]. To improve its stabilization and performance, different strategies have been implemented. These strategies include synthetizing it in monocrystalline and polycrystalline structures [149], the ZnO surface coating to increase the structural stability and the conductivity (the corresponding battery performance is shown in Figure 6a) [150], or the use of Ag-Sn dual-modification to promote structural stability (Figure 6b) [151]. $LiNi_{0.8}Co_{0.1}Mn_{0.1}O_2$ has been also doped with reduced graphene oxide (rGO) [152] and with a $La_2Zr_2O_7$ coating for Zr doping (cycling behavior shown in Figure 6c) [153]. Finally, other modification strategies for the same active material include surface modification after washing by $H_3BO_3$ [154], 3D carbon network using 6-amino-4-hydroxy-2-naphthalenesulfonic acid (AHNS)-functionalized rGO and carbon nanotubes (CNTs) [155], partially substitution of Co for Fe [156], and adding ethylene glycol (EG) and surfactant polyvinylpyrrolidone (PVP) [157].

A further variety of related active materials include $LiNi_{0.5}Co_{0.2}Mn_{0.3}O_2$ coated with an oligomer additive [158] and produced by drying methods (freeze drying and vacuum drying) through the solvothermal method [159], $LiNi_{0.6}Co_{0.2}Mn_{0.2}O_2$ with Zr-based dual modification [160], $LiNi_{0.6}Co_{0.2}Mn_{0.2}O_2$ synthesized via solid reaction assisted with a plasma milling pretreatment [161], $0.5Li2MnO3.0.5LiMn_{1/3}Ni_{1/3}Co_{1/3}O_2$ [162], $LiNi_{0.83}Co_{0.10}Mn_{0.07}O_2$, [163], $LiNi_{0.83}Co_{0.12}Mn_{0.05}O_2$/graphite–$SiO_x$ [164], $LiNi_{0.87}Co_{0.1}Al_{0.03}O_2$ with the LBP coating layer [165], $LiNi_{0.88}Co_{0.06}Mn_{0.03}Al_{0.03}O_2$ [166], $LiNi_{0.88}Co_{0.09}Al_{0.03}O_2$ coated with $Al_2O_3$

layer [167], LiNi$_{0.90}$Co$_{0.05}$Mn$_{0.05}$O$_2$ with lithium tungsten oxide (LWO) coating layer [168] and Sn-doping [169], Li$_{1.02}$Ni$_{0.05}$Mn$_{1.93}$O$_4$ [170], Li$_{1.05}$Mn$_{1.95}$-xNi$_x$O$_4$ (0≤ x ≤ 0.08) by a solution combustion method [171], LiMn$_{0.5}$Fe$_{0.5}$PO$_4$ cathode coated of Li$_3$VO$_4$, and carbon [172], Li$_2$ZrO$_3$-Li$_3$V$_2$(PO$_4$)$_3$/C composites [173], among others, have been also explored with great potential for cathode electrode aiming to increase battery performance and stability.

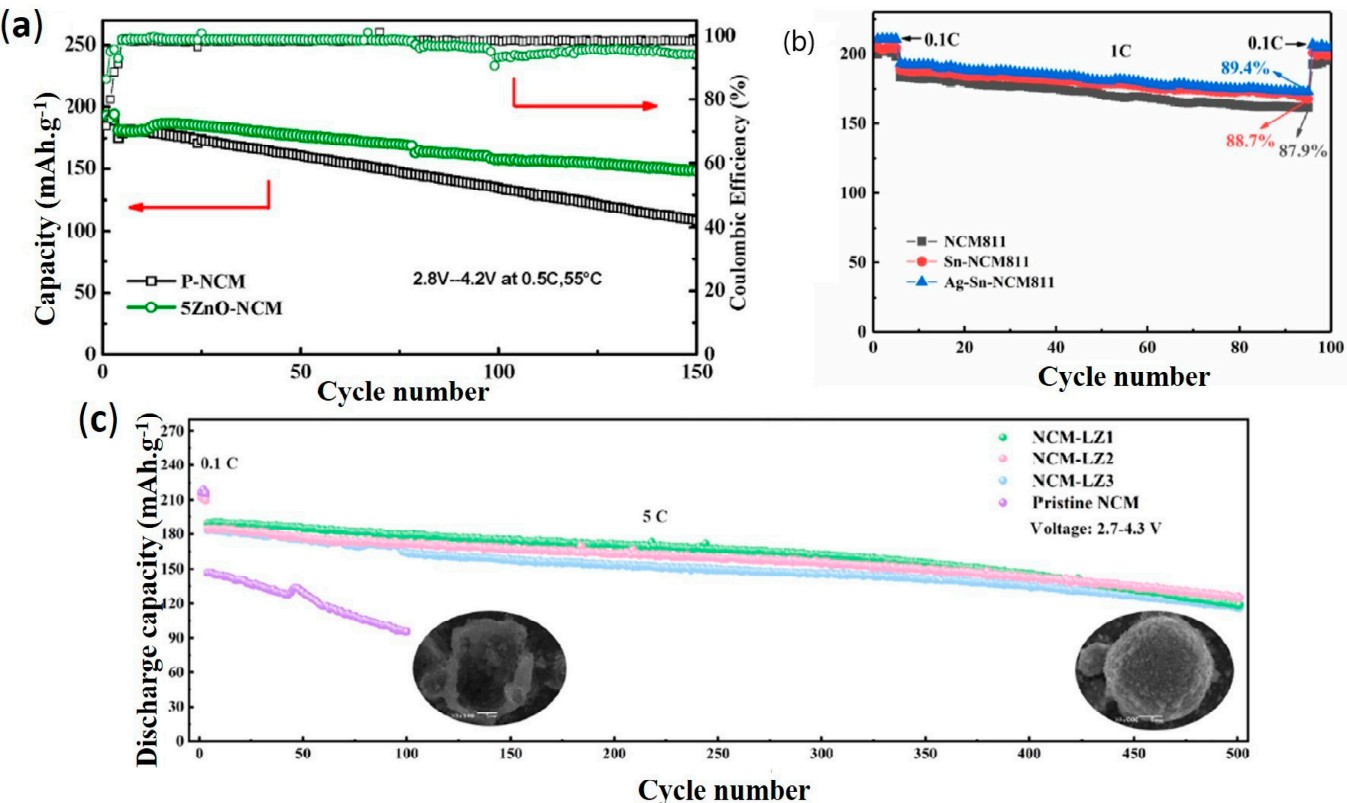

**Figure 6.** (**a**) Cyclic performance at 55 °C of NCM and 5ZnO-NCM [150], (**b**) Cycling performance of NCM811, Sn-NCM811, and Ag-Sn-NCM811 (1 C = 200 mA·g$^{-1}$, at the range of 2.8–4.3 V) [151], and (**c**) Cycling performance of the a La$_2$Zr$_2$O$_7$-coated NCM samples at 5 C [153].

In addition, active cathode materials based on single crystal LiNi$_{0.83}$Co$_{0.10}$Mn$_{0.07}$O$_2$ (SC-NCM) have been developed to improve thermal stability, offering a reversible capacity of 167.0 mAh·g$^{-1}$ [163]. LiNi$_{0.88}$Co$_{0.06}$Mn$_{0.03}$Al$_{0.03}$O$_2$ has been synthesized and modified by Zr doping and LiBO$_2$ coating to reach a discharge specific capacity of 211.7 mAh·g$^{-1}$ at 0.1 C [166]. A composition of Li$_{1.17}$Ni$_{0.21}$Mn$_{0.54}$Co$_{0.08}$O$_2$, with low cobalt content, synthetized by the co-precipitation method, reaches a specific capacity of 250 mAh·g$^{-1}$, the electrochemical behavior being governed by Ni$^{2+}$/Ni$^{4+}$ and Co$^{3+}$/Co$^{4+}$ redox couples [174]. Li$_{1.2}$Ni$_{0.182}$Co$_{0.08}$Mn$_{0.538}$O$_2$ has been coated with PVP-bridged γ-LiAlO$_2$ nanolayers to improve the rate capability and cycling stability, the rate capability being 177.0 mAh·g$^{-1}$ at 5 C [175].

Another interesting active material is Li$_{1.2}$Mn$_{0.54}$Co$_{0.13}$Ni$_{0.13}$O$_2$, which doped with carbon and oriented following a {010} plane allows improving Li$^+$ diffusion and reaching a reversible capacity of 276 mAh·g$^{-1}$ [176]. Another strategy for this active material is to modify the surface using La–Co–O compounds [177]. Finally, this material has been also synthetized to improve the kinetics and reversibility of transition-metal (TM) ion migration, leading to an ultrahigh energy efficiency at 1 C (90.6%) and high capacity (>200 mAh·g$^{-1}$) [178].

Other cathode active materials without lithium in their structure have been studied as an alternative of the conventional lithium cathode active materials. In this case, the source

of lithium ions is the lithium metal used as anode. The electrode material $NaVMoO_6$ with a layered structure has been found with the highest valence states for vanadium (+5) and molybdenum (+6), which are suitable for cathode active material for LIBs, their structure and performance being shown in Figure 7a,b, respectively [179].

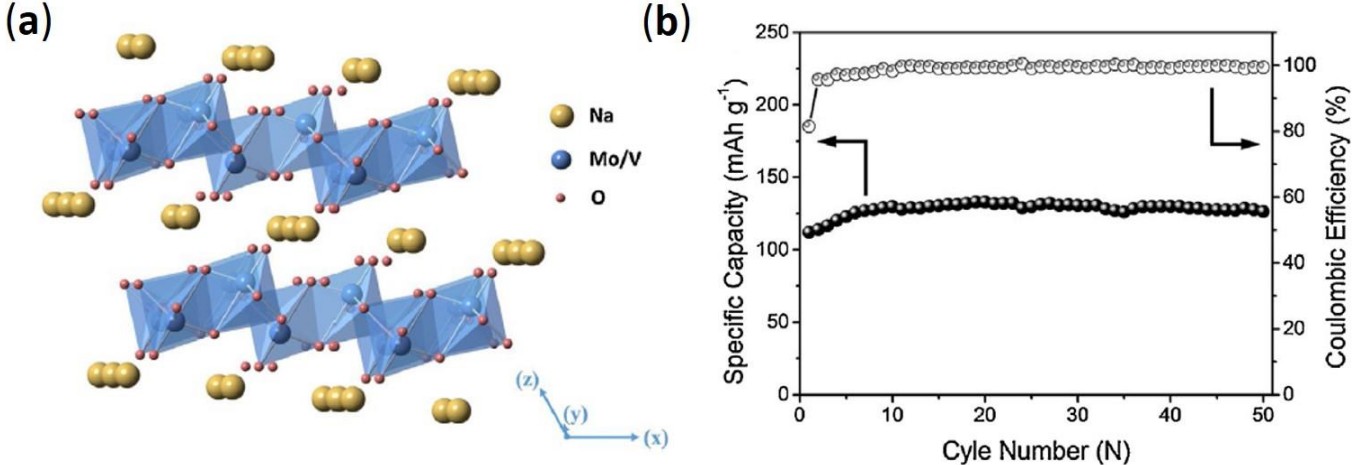

**Figure 7.** (**a**) Schematic structure and (**b**) battery performance for $NaVMoO_6$ [179].

In addition, a ferrocene-based MOF, iron (III) 1,1′-ferrocenedicarboxylate ($Fe_2(DFc)_3$), has been synthesized and tested for cathode applications with the specific capacity of $172 \ mAh \cdot g^{-1}$ at $50 \ mA \cdot g^{-1}$ [180].

## 4. Materials for Separators/Electrolytes

### 4.1. Separator Membrane

An essential component of electrochemical devices is the separator. This component is placed between the two electrodes, and its main function is to prevent the short circuit of the system. Its specific function depends on the type of application being, in the case of lithium-ion batteries, the means of transferring lithium-ions between the electrodes during the charge/discharge process [181].

There are different types of separators, such as microporous membranes, nonwoven membranes, electrospun membranes, membranes with external surface modification, composite membranes, and polymer blends [182].

The separator is composed by a porous membrane wetted with an organic electrolyte solution, the main characteristics required for battery applications being [32,182]:

- Low thickness (<25 μm) and good permeability;
- Porosity > 50% and pore size below <1 micron;
- Excellent wettability: absorption and retention of electrolytes;
- Chemical and dimensional stability;
- Good thermal stability and excellent mechanical properties.

Different separator types are produced using different processing techniques, ranging from solvent casting to electrospinning, and different polymers have been used, including polyethylene (PE), polypropylene (PP), poly (vinyl chloride) (PVC), poly(ethylene oxide) (PEO), and poly (vinylidene fluoride) (PVDF), among others, commercial separators being based on polyolefins (PP and PE). Regardless of the separator type, its thickness should be lower than 25 μm, porosity higher than 40% with an average pore size lower than 1 μm and stable at temperatures above 150 °C [182].

Advances in the area of separators are focused on the development of advanced porous membranes based on environmental friendly materials, improving thermal and safety properties, increasing the wettability of polyolefins polymers, and inhibiting dendrites growth [183], among others. Table 3 summarizes representative recent works in this field.

**Table 3.** Recent advances on Li-ion battery separators based of different materials with main properties, goals and achievements for improving separator characteristics.

| Materials | Electrolyte Solution | Porosity and Uptake (%) | Conductivity (mS·cm$^{-1}$) and Capacity (mAh·g$^{-1}$) | Main Goal/Achievement | Ref. |
|---|---|---|---|---|---|
| Polyimide (PI) with poly(amic acid) (PAA) | 1.0 M LiPF$_6$ (ethylene carbonate (EC)/ diethyl carbonate (DEC) (1:1 by weight)) with 5% of fluoroethylene carbonate (FEC) | 89.1/- | 1.79/1.930 mAh | Improved mechanical strength | [184] |
| Polyimide (PI) | 1.0 M LiPF$_6$ (EC/dimethyl carbonate (DMC) (1:1 in vol.)) | -/- | -/- | Improved thermal stability | [185] |
| PE with phenolic resin (AF) | 1mol L$^{-1}$ LiPF$_6$ (DMC/EC (1:1 in vol.)) | 57/228 | 0.6/119 | Improved thermal stability and electrochemical properties | [186] |
| Untreated Al$_2$O$_3$/PE | 1.0 M LiPF$_6$ (EC/DEC (1:1 by weight)) with 5% of Fluoroethylene carbonate (FEC) | -/- | 0.39/140@0.2C | Good wettability, high thermal stability, and good electrochemical performance | [187] |
| Polyethyleneimine (PEI)/dopamine coating layer in PP separator. | 1.0 M LiPF$_6$ (EC/DMC (1:1 in vol.)) | -/144 | 0.58/128 | High electrolyte uptake | [188] |
| PVDF containing titanium dioxide (TiO$_2$) and graphene oxide (GO) | Commercial electrolyte based LiPF$_6$ | 86.50/494 | 4.87/- | High electrolyte uptake | [189] |
| Polyimide (PI) with ZSM-5 zeolite filler | 1.0 M LiPF$_6$ (EC/DMC (1:1 in vol.)) | 61/260 | 1.04/133@2C | Enhanced wettability and electrolyte uptake | [190] |
| Poly(aryl ether sulfone) (PES) and poly(vinylidene fluoride) (PVDF) | 1.0 M LiPF$_6$ (EC/ethylmethyl carbonate (EMC)/DMC (1:1:1 in vol.)) | -/595 | 1.69/162.8 | Enhanced wettability and high ionic conductivity | [191] |
| Poly(vinylidene fluoridehExafluoropropyl-ene) (PVDF-HFP)/poly-mphenyleneisophthalamide (PMIA) | 1.0 M LiPF$_6$ (EC/DEC/EMC (1:1:1 in vol.)) | 94.28/~900 | -/- | Good electrolyte affinity and enhanced interfacial compatibility | [192] |

**Table 3.** *Cont.*

| Materials | Electrolyte Solution | Porosity and Uptake (%) | Conductivity (mS·cm$^{-1}$) and Capacity (mAh·g$^{-1}$) | Main Goal/Achievement | Ref. |
|---|---|---|---|---|---|
| PE with ammonium persulfate (APS) coating | 1.0 M LiPF$_6$ (EC/EMC (3:7 in vol.)) with 2 wt % vinylene carbonate as an additive | -/- | 0.36/~170 | High lithium-ion migration and ionic conductivity | [193,194] |
| PVDF/13X zeolite particles | 1.0 M LiPF$_6$ (EC/DEC/DMC (1:1:1 in vol.)) | 76/475 | ~1/144.14 | Excellent ionic conductivity | [195] |
| PAN@PVdF-HFP | - | -/- | 1.2/170@0.1C | Excellent cycling performance, good rate capability | [196] |
| PE with controllable polyamine (PAI) layer | 1.0 M LiPF$_6$ carbonate solution | 60/- | -/- | Enhanced safety | [197] |
| PVDF coated with ZnO | - | 85.1/352 | 2.3/148@1C | High safety in high temperature. | [198] |
| Thin layer of low-density polyethylene microspheres onto a commercial porous PP | 1.0 M LiPF$_6$ (EC/DMC/EMC (1:1:1 in vol.)) | -/- | 0.30/158 | Rapid thermal shutdown at elevated temperature (≈110 °C) | [199] |
| Aramid nanofiber/ bacterial cellulose | 1.0 M LiPF$_6$ (EC/DMC/DEC (1:1:1 in vol.)) | 83.9/- | 12.54/157 | Excellent tensile strength and ionic conductivity. | [200] |
| Poly (L-lactic acid) (PLLA) | 1.0 M LiPF$_6$ (EC/DMC (1:1In vol.)) | ≈72/350 | 1.6/93@1C | Environmentally friendly separator | [201] |
| Cellulose/PVDF-HFP with TiO$_2$ | 1.0 M LiPF$_6$ (EC/DMC (1:1 in vol.)) | 86/403 | 1.68/103.8@8C | Excellent thermal stability and high ion conductivity | [202] |
| Silk fibroin | 1.0 M LiPF$_6$ (EC/DMC (1:1 in vol.)) | 86/350 | 2.2/131.3 @8C | Environmentally friendly separator | [203] |

Stability parameters, such as thermal and mechanical characteristics, are the easiest to achieve, as the properties of the separator components are well known and easily combined. Polyimide (PIs) is widely used in the separator field due to the improved thermal stability and different separators based on this polymer are being developed, such as PI nonwovens with diphenyl phosphate (DPhP) as plasticizer [204], PI with organic montmorillonite (OMMT) [205], a three-dimensionally ordered microporous polyimide (3DOM PI) separator developed by micropatterning [206], coating of silicon nitride on both sides of polyimide separator [207], and a PVDF-HFP/PI side-by-side bicomponent electrospun separator [208]. In addition to PI, other polymers such as poly(aryl ether benzimidazole) (OPBI) [209], polydopamine-coated poly(m-phenylene isophthalamid) membrane [210], and poly(phenylene sulfide) [211] are used due to their high thermal resistance.

The affinity of the separator with the electrolytes is also an important parameter, which is mainly associated with the wettability and uptake of the membranes. In order to improve electrolyte wettability and thermal stability, new separators based on heat-resistant polyphenylene sulfide (PPS) fibers and cellulose fibers (CFs) were fabricated via a facile papermaking process, and their performance is shown in Figure 8a [212]. The wettability of PE separator membranes can be improved by the coating of $Al_2O_3$ on the membrane, leading also to a reduction of the interfacial resistance [187]. Furthermore, in the same separator type and with the objective of enhancing thermal runaway, melamine-based porous organic polymer (POP) coatings [213] were implemented. In fact, different materials have been used as a layer on the surface of polyolefin separators to improve its wettability, including polyethyleneimine (PEI)/dopamine coating layer [188], $Al_2O_3$ layers in electrospun PVDF nanofibers [193], ammonium persulfate (APS) coating [194], a phenolic resin (AF) layer with immersion in situ reaction [186], a thin layer of low-density polyethylene microspheres [199], polyamine (PAI) containing natural clay nanorods (attapulgite, ATP) [197], and polyvinyl alcohol (PVA) [214], $TiO_2$ [215], $Al_2O_3$ [216], UV-induced graft with polar methyl acrylate (MA) [217], and polyimide (PI)-$SiO_2$ layers [218].

The ionic conduction of the separators is a direct consequence of their wettability and uptake, as the electrolyte plays a key role on the electrochemical properties of the system. In this regard, composite separators are intensively used, and different combinations of polymer matrix and specific fillers are being developed, such as, boehmite/polyacrylonitrile (BM/PAN) [219], 9,10-dihydro-9- oxa-10-phosphaphenanthrene-10-oxide (DOPO) into polyacrylonitrile (PAN) [220], PVDF containing titanium dioxide ($TiO_2$) and graphene oxide (GO) [189], PVDF with 13X zeolite [195] and PVDF with modacrylic and $SiO_2$ [221], poly-acrylonitrile (PAN)/helical carbon nanofibers(HCNFs)@PVDF/UiO-66 composite [222], cellulose/Poly (vinylidene fluoride-hexafluoropropylene) membrane with titania nanoparticles [202], polyimide (PI) with ZSM-5 zeolite as filler [190] and PVDF with titanium hydroxide (Ti(OH)$_x$) [223], polyethylene terephthalate (PET) combined with inorganic zirconia ($ZrO_2$) [224], silica-coated expanded polytetrafluoroethylene separator [225], poly(vinyl alcohol) (PVA) with $ZrO_2$ nanoparticles [226], poly(vinyl alcohol) (PVA) with submicron spindle-shaped $CaCO_3$ [227], poly(vinyl alcohol)/melamine composite nanofiber membrane containing LATP nanocrystals [228], and poly(m-phenylene isophthalamide) (PMIA) with $SiO_2$ nanoparticles [229], among others, mainly with the main focus on improving the electrochemical properties. In particular, separators based on PVDF coated with ZnO have been developed with higher ionic conductivity (2.261 mS·cm$^{-1}$), high porosity (85.1%), favorable electrolyte wettability (352%), and lower interfacial impedance (220 Ω) [198].

The future trends on the separator technology include the development of more complex and multifunctional membranes, particularly with the increasing of the devices' safety, by adding shutdown functions to the separators [199]. Novel separator architectures based on three layers were developed combining electrospinning and electrospraying techniques and composed of PI and poly(amic acid) ammonium salt (PAAS) solution with inorganic nanoparticles ($SiO_2$ or $Al_2O_3$) [185]. Another three-layer separator was produced based on poly (ethylene-co-vinyl acetate) (EVA)/polyimide (PI)/EVA (PIE) with high thermal stability and a shutdown function [230].

With the increase of the global awareness on environmental issues, the development of "greener" separators for batteries has also gained relevance in recent years, with the application of biopolymers as host, and less hazardous fillers in order to reduce environmental impacts. A biopolymer widely used in battery separator membranes is cellulose. In order to improve its performance, the "Trojan Horse" camouflage strategy was used, which consists of preparing positively charged lignosulfonate–polyamide–epichlorohydrin complex (LPC) nanoparticles that are incorporated into the separator. This strategy shows exceptional electrolyte wettability and rate capability as shown in Figure 8b [231]. Other works in this field include composites based on bacterial cellulose nanocrystals (BCNCs) with polyether block amide (PEBAX) [232], zeolitic imidazolate framework-67 (ZIF-67) on the surface of cellulose nanofibers (CNFs) [233], poly(vinylidene fluoride-hexafluoropropylene)/cellulose/carboxylic titanium dioxide (PVDF-HFP/cellulose/C-TiO2) composites [234], aramid nanofiber (ANF)/bacterial cellulose (BC) [200], cellulose nanofibrils (CNFs) reinforced pure cellulose paper (CCP) [235], and Lyocell fibrillated fibers [236], all showing exceptional electrochemical performance and rate capability (Figure 8c for ANF/BC separator). A novel PVDF/triphenyl phosphate (TPP)/cellulose acetate (CA) separator membrane was fabricated by electrospinning, and this membrane shows high porosity, improved thermal stability, superior electrolyte wettability, improved flame resistance, excellent electrochemical properties, and cycle stability when compared to the commercial separators (Figure 8d) [237]. Silk fibroin membranes prepared by salt leaching are also an excellent candidate for this green transition, due to its high porosity and uptake, which leads to excellent battery performances at a wide range of discharge rates [203]. The experimental studies are frequently accompanied with theoretical and simulation models to better understand the involved physical–chemical phenomena [238].

*4.2. Solid Polymer Electrolytes*

Solid polymer electrolytes (SPE) are a solvent-free salt solution in a polymer host, which can be considered solid in the macroscopic scale [37]. They are a key component in the operation of solid-state batteries, as they allow the removal of the liquid electrolytes from the system, improving their safety. However, some major drawbacks are causing this technology to lag behind, namely the low ionic conductivity achieved and the difficult interfacial interaction between the electrodes and the electrolyte [239,240]. A good SPE in LIBs must have some basic properties, such as good ionic conductivity ($>10^{-4}$ S·cm$^{-1}$), good interfacial compatibility with the electrodes, high lithium-ion transference number, and good mechanical and thermal stability. The perfect balance among all these characteristics is difficult to achieve but will allow producing suitable SPE for application at large scale in the next generation of solid-state batteries [38].

The field of SPEs in lithium-ion batteries is strongly growing in recent years, with more than 500 publications in 2020 according to the Web of Science. This highlights the relevance of this thematic and the efforts made to find the most suitable materials for application in SPEs. Some of the most relevant studies are focusing on the resolution of issues that include the growth of lithium dendrites [241] and electrode/electrolyte interface [242], as well as enhancing the multifunctionality of the SPE, with the addition of battery shutdown functions or self-healing ability. Table 4 summarizes and outlines the recent works in the SPE field.

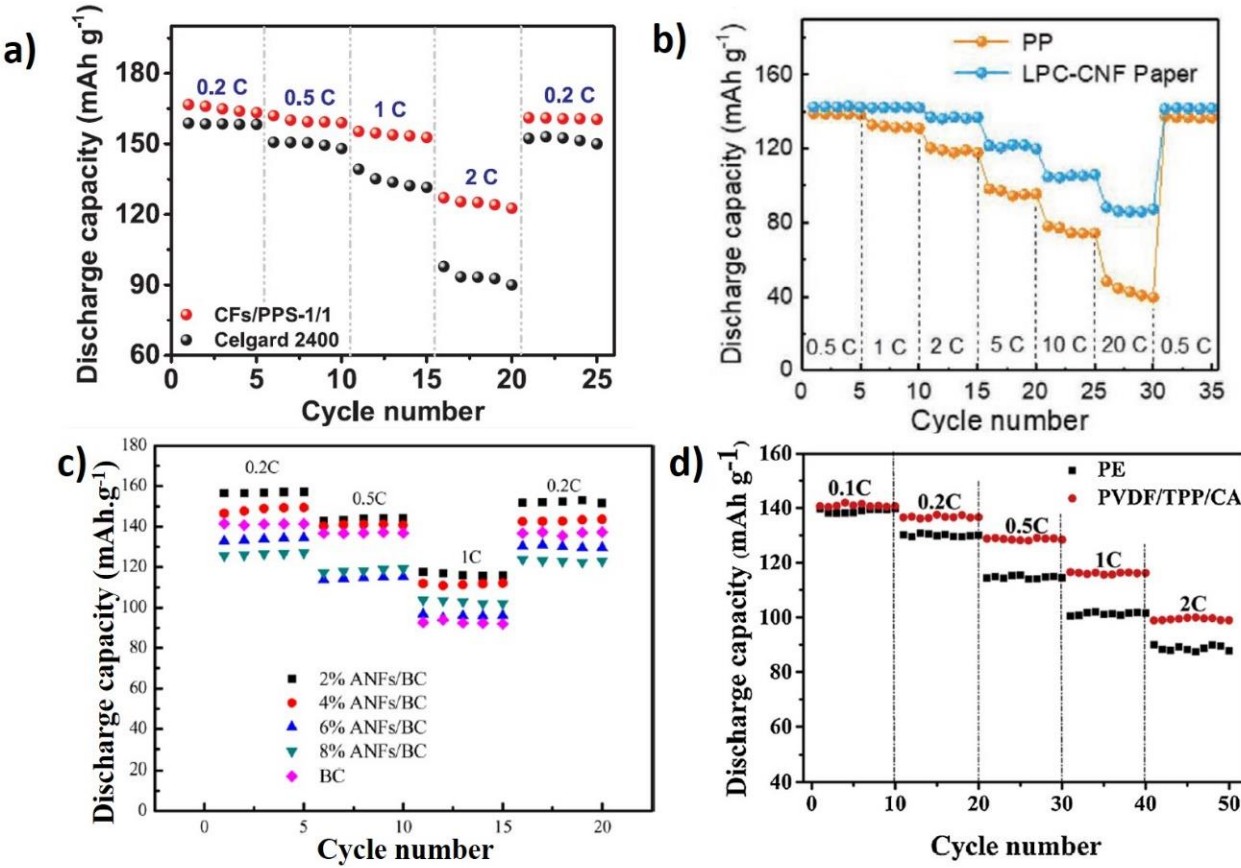

**Figure 8.** (**a**) Discharge C-rate capabilities of cells with Celgard 2400 separator and CFs/PPS-1/1 separator [212], (**b**) Rate capability of LFP half cells with PP and LPC-CNF paper separators at different current densities [231], (**c**) Rate capability of LFP half-cells with BC and ANFs/BC separators [200], and (**d**) Rate capabilities of LFP half cells using PE and PVDF/TPP/CA membranes [237].

**Table 4.** Recent advances in solid polymer electrolyte (SPE) technology for Li-ion battery applications.

| Polymer Matrix | Fillers | Method | Ionic Conductivity ($S \cdot cm^{-1}$) | Discharge Capacity ($mAh \cdot g^{-1}$) | Main Features | Ref. |
|---|---|---|---|---|---|---|
| PEO | Carbon quantum dots, LiTFSI | Doctor blade | $2.2 \times 10^{-4}$ (25 °C) | 160.4 (C/10) | Lithium dendrite suppression | [30] |
| PEO | GO, [Bmim][DCA] | Solvent casting | $1 \times 10^{-4}$ (25 °C) | 156.2 (C/10) | Lithium dendrite suppression | [243] |
| PEO | LiTFSI $SiO_2$ | Solvent casting | $9.32 \times 10^{-5}$ (30 °C) | 166.9 (C/10) | Lithium dendrite suppression | [244] |
| PEO | Nonwoven glass fiber, LiTFSI | Solvent casting | $1.2 \times 10^{-4}$ (20 °C) | 128 (C/10) | Lithium dendrite suppression | [245] |
| PEO, PVDF | LiTFSI | Solvent casting | $2.46 \times 10^{-9}$ (30 °C) | 157.5 (C/5) | Lithium dendrite suppression | [246] |

**Table 4.** *Cont.*

| Polymer Matrix | Fillers | Method | Ionic Conductivity (S·cm$^{-1}$) | Discharge Capacity (mAh·g$^{-1}$) | Main Features | Ref. |
|---|---|---|---|---|---|---|
| PCL-PPC-PCL | LiTFSI | Solvent casting | $3 \times 10^{-5}$ (30 °C) | 142 (C/20) | Lithium dendrite suppression | [247] |
| PVDF | LiTFSI, LLZTO | Solvent casting | $1.16 \times 10^{-3}$ (80 °C) | 151 (C/5) | Lithium dendrite suppression | [248] |
| PVDF | LiTFSI, Pyr$_{13}$TFSI | Solvent casting | $1.23 \times 10^{-3}$ (25 °C) | 158.2 (1C) | Lithium dendrite suppression | [249] |
| PVDF-CA | Montmorillonite, LiTFSI | Solvent casting | $3.40 \times 10^{-4}$ (25 °C) | 112 (C/2) | Lithium dendrite suppression | [250] |
| PVDF-HFP | LLZO, IL | Solvent casting | $6.3 \times 10^{-3}$ (20 °C) | 164 (C/2) | Lithium dendrite suppression | [251] |
| PEO | LiTFSI, UiO-66-NH$_2$@SiO$_2$ | Hot pressing | $8.1 \times 10^{-6}$ (60 °C) | 151 (C/10) | Lithium dendrite suppression | [252] |
| TMPTA-TEGDME-PEO | LiTFSI | UV curing | $4.36 \times 10^{-4}$ (30 °C) | 157.8 (C/10) | Lithium dendrite suppression | [253] |
| PEO | LiTFSI, Pyr$_{14}$TFSI, LLZO | UV curing | $5.0 \times 10^{-4}$ (60 °C) | - | Lithium dendrite suppression | [29] |
| PVDF | LiClO$_4$, LLTO | Tape casting | $4.7 \times 10^{-4}$ (25 °C) | 139 (C/5) | Lithium dendrite suppression, interfacial contact | [241] |
| PAN nanofibers, PDMS, PEO | LiTFSI | Solvent casting | $1.2 \times 10^{-3}$ (60 °C) | 151.7 (C/5) | Lithium dendrite suppression, interfacial contact | [254] |
| PEO | LiTFSI Zn(BEH$_2$) | Solvent casting | $1.1 \times 10^{-5}$ (30 °C) | 145 (C/10) | Lithium dendrite suppression, interfacial contact | [255] |
| PVDF-HFP | LiTFSI, LGPS | Solvent casting | $1.8 \times 10^{-4}$ (30 °C) | 158 (C/20) | Lithium dendrite suppression, interfacial contact | [256] |
| PEG | LiTFSI, HDIt | Cross-linked copolymerization | $6.51 \times 10^{-5}$ (25 °C) | 162 (C/10) | Interfacial contact | [242] |

**Table 4.** *Cont.*

| Polymer Matrix | Fillers | Method | Ionic Conductivity (S·cm$^{-1}$) | Discharge Capacity (mAh·g$^{-1}$) | Main Features | Ref. |
|---|---|---|---|---|---|---|
| PEO | LiFSI, LiPSTFSI | Hot pressing | $3.7 \times 10^{-5}$ (70 °C) | 150 (C/20) | Interfacial contact | [257] |
| PEO | LiPCSI | Solvent casting | $7.33 \times 10^{-5}$ (60 °C) | 141 (C/10) | Interfacial contact | [258] |
| PEO | Li$_3$N | Doctor blade | - | 160 (C/10) | Self-healing | [259] |
| HCP-UPyMA, PEGMA | LiTFSI | UV copolymerization | $8.95 \times 10^{-5}$ (30 °C) | - | Self-healing | [260] |
| PVT | [EMIM][TFSI], LiTFSI | Solvent casting | $1.26 \times 10^{-4}$ (25 °C) | 145 | Self-healing | [261] |
| PEGA | LiTFSI, Bis(2-methacryloyloxyethyl) Disulfide, 1,2-Bis (ureidoethylen-emethacrylate) Hexamethylene | RAFT polymerization | $7.28 \times 10^{-6}$ (30 °C) | 140.5 (C/10) | Self-healing | [262] |
| PEG600 | Phosphorous and silicon-containing monomers, LiTFSI | Solvent casting | $2.98 \times 10^{-5}$ (25 °C) | 142.0 (C/10) | Flame retardancy | [263] |
| PAES-g-PEG | PYR14-TFSI, LiTFSI | Solvent casting | $8.9 \times 10^{-4}$ (40 °C) | 138 (C/10) | Battery stability | [264] |
| PEO | ZIF-8 | Solvent casting | $2.2 \times 10^{-5}$ (30 °C) | 111 (C/2) | Battery stability | [265] |
| ETPTA-PVDF-HFP | - | UV curing | $9 \times 10^{-4}$ (25 °C) | 150 (C/5) | Ionic conduction | [266] |
| PEO | UiO-66, LiClO$_4$ | Solvent casting | $4.8 \times 10^{-5}$ (25 °C) | 148 (C/10) | Ionic conduction | [267] |
| PEO | HACC-TFSI, LiTFSI | Solvent casting | $1.77 \times 10^{-5}$ (30 °C) | 161.3 (C/5) | Ionic conduction | [268] |
| PEO | LiTFSO, Mesoporous silica | Solvent casting | $4.3 \times 10^{-4}$ (60 °C) | 150.3 (C/10) | Ionic conduction | [269] |
| PVDF-HFP | LiTFSI, Pyr$_{13}$TFSI, P(MMA-co-VIm$_{(1}$O$_2$)) (TFSI) | Solvent casting | $5.1 \times 10^{-4}$ (25 °C) | 102 (C/10) | Ionic conduction | [270] |
| PMMA | BaTiO$_3$, LiPF$_6$ | Solvent casting | $3.9 \times 10^{-4}$ (70 °C) | - | Ionic conduction | [271] |
| PVDH-HFP | [Bmim][SCN] | Doctor blade | $1.5 \times 10^{-4}$ (25 °C) | 148 (C/8) | Ionic conduction | [272] |
| PEO | LiTFSI | Ultrasonic treatment, solvent casting | $3.2 \times 10^{-4}$ (25 °C) | - | Ionic conduction | [273] |
| PEO | LiTFSI, LLZAO | Solvent casting | $2.51 \times 10^{-4}$ (25 °C) | 165.9 (C/5) | Ionic conduction | [274] |
| PVO | LiTFSI | Solvent casting | $1.36 \times 10^{-6}$ (25 °C) | - | Ionic conduction | [275] |

| Polymer Matrix | Fillers | Method | Ionic Conductivity (S·cm$^{-1}$) | Discharge Capacity (mAh·g$^{-1}$) | Main Features | Ref. |
|---|---|---|---|---|---|---|
| Chitosan, PEG | LiClO$_4$ | Solvent casting | $4.56 \times 10^{-4}$ (25 °C) | - | Environmentally friendly | [276] |
| I-Carrageenan | LiCl | Solvent casting | $5.33 \times 10^{-3}$ (25 °C) | - | Environmentally friendly | [277] |
| Pectin, Guar gum | LiTFSI | Solvent casting | $1.59 \times 10^{-4}$ (25 °C) | - | Environmentally friendly | [278] |
| PAA | Silica nanoparticles | Free radical polymerization | $1.29 \times 10^{-2}$ (25 °C) | - | Environmentally friendly | [279] |

The lithium dendrite suppression can be a major breakthrough for the use of lithium metal batteries in their full potential, without safety problems. According to that, several studies have been carried out in order to solve this issue. The production of a sandwiched PVDF/LLTO by tape casting proved to be effective for this purpose, as the use of different LLTO concentrations in each layer allows for high ionic conductivities and good interfacial compatibility. This leads to batteries with good cycling capacities and excellent lifetimes [241]. Similar results are reported for LLZO nanofibers [251]. LLZO can also be combined with PEO and Pyr$_{14}$TFSI to reinforce commercial Celgard® separators, leading to an SPE with excellent dendrite suppression ability. The formation of dense Li depositions in the interface with lithium metal is observed, due to a synergistic electro-chemo-mechanical effect observed in LLZO composite layer (Figure 9) [29]. The addition of 3D SiO$_2$ particles into a PEO matrix offers a uniform dispersion of high conductive interfaces, improving the performance of the battery [244]. The 3D cross-linked network formed by a mixture of PEO, GO, and [Bmim][DCA] provided a fast ion transport channel in the SPE structure, leading to high lithium transference number and high ionic conductivity even at room temperature, simultaneously suppressing the lithium dendrite growth [243]. Ionic liquids such as Pyr$_{13}$TFSI lead to a viscoelastic electrode/electrolyte interface, which reduces the impedance of the battery and avoids lithium dendrites [249]. The use of MOFs, such as UiO-66-NH$_2$@SiO$_2$, leads to a network of high conductivity channels in the SPE structure, which improves its electrochemical properties and reduces the lithium dendrite formation [252]. By adding Zn(BEH$_2$) at the SPE, a LiZn alloy interface is formed in the anode, which suppresses dendrite penetration, simultaneously increasing the ionic conductivity [255]. The highly conductive LGPS salts act as stabilizers, particularly when combined with fluorinated polymers such as PVDF-HFP, due to the LiF layer formed between the electrodes and the electrolyte, which suppresses lithium propagation [256]. The addition of carbon quantum dots into a PEO matrix leads to an increase in the mechanical stability of the SPE, particularly at the level of stretching and puncture resistance, without compromising the battery performance [30]. The use of nonwoven glass fiber to reinforce the PEO matrix significantly improves the mechanical properties of the SPE with storage moduli up to 1 GPa. This allows not only the lithium dendrite suppression but also the possibility of increasing the load of other fillers, leading to higher ionic conductivities and cycling performances without compromising the mechanical properties [245].

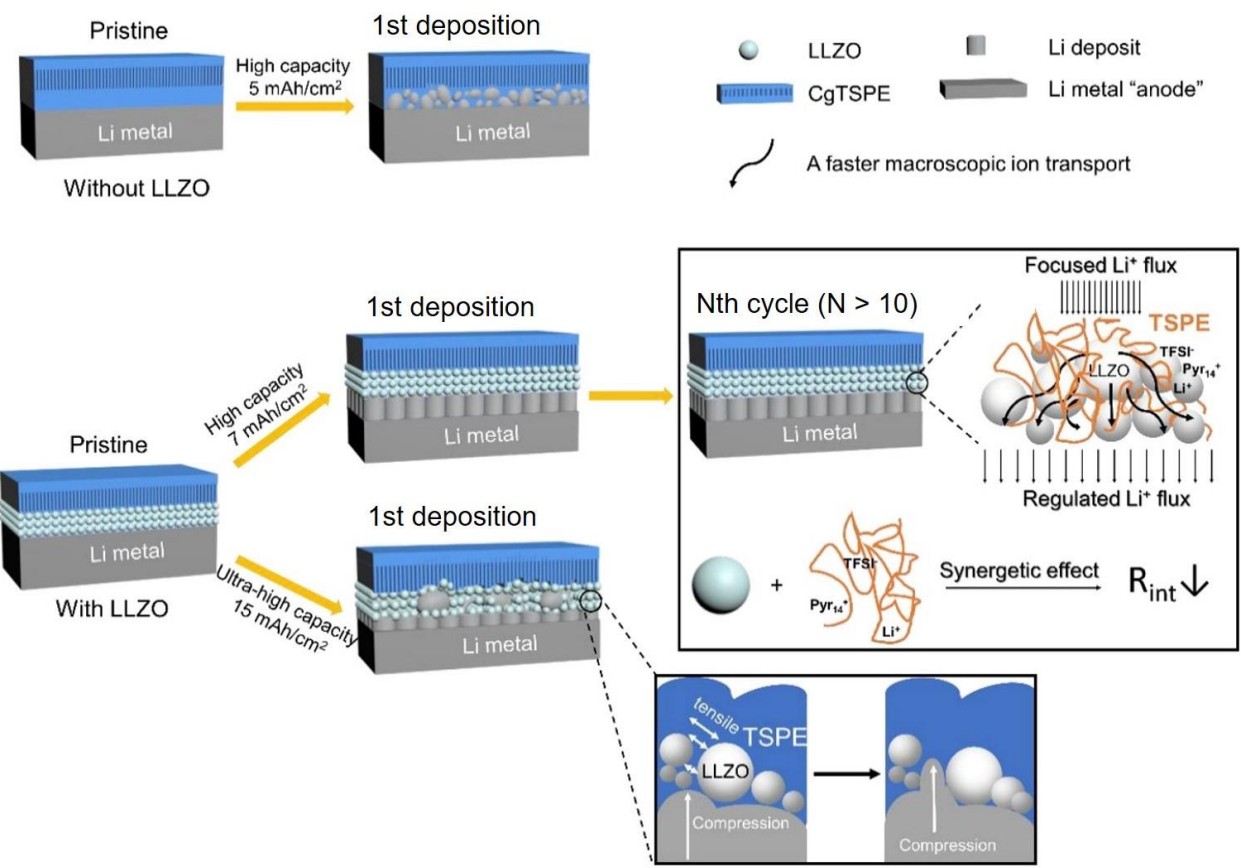

**Figure 9.** Effect of an LLZO layer in the suppression of lithium dendrites growth [29].

Another strategy to suppress lithium dendrites is the combination of different kinds of polymers. By adding PAN electrospun nanofibers to a blend of PDMS and PEO, it is possible to reduce the crystallization of PEO and increase the mechanical properties of the SPE, which prevents it from being punctured by dendrites [254]. The addition of a PEO/LiTFSI solution in an electrospun PVDF nanofiber matrix also reduces the PEO crystallization due to the increase in the contact area with the nanofibers. This increases the mechanical strength and the cycling stability of the SPE [246]. Similar results can also be achieved by using a polyester-based triblock structure [247]. By combining different kinds of polymers in regular-random network structures, it is possible to regulate the lithium deposition by facilitating the formation of a regular and robust SEI in lithium metal [253]. Mechanical properties improvement can also be achieved by creating polymer blends between PVDF and cellulose acetate. Simultaneously, the addition of an inorganic filler (montmorillonite) accelerates the dissociation process of LiTFSI, allowing for higher ion transport rates [250]. The development of a hybrid polymer-in-ceramic composite electrolyte is an interesting approach, as it combines the advantages of high ionic conductivity in ceramic electrolytes, and the mechanical properties of the polymer electrolytes; furthermore, it can simultaneously suppress the lithium dendrites [248].

In many cases, the suppression of lithium dendrites is strongly associated with the improvement of the interfacial contact between the electrodes and the electrolyte [254–256]. The interfacial compatibility can be enhanced by combining different functional units, as PEG and HDIt in one SPE, in which its properties can be easily tuned by changing the ratio between the functional units [242]. The use of additive containing solid lithium-ion conductors, where lithium salts are combined with polymers such as PS, proved to reduce the interfacial resistivity between the electrolyte and the electrodes, improving battery performance [257]. The substitution of the conventional lithium salts by LiPCSI reduces

the glass transition temperature of the SPE, thus increasing the mobility of the polymer chains and improving compatibility with metallic lithium anodes [258].

An interesting approach that attracted great interest in recent years is the study of the self-healing capacity in SPEs. This application has the potential to solve simultaneously the interfacial and lithium dendrite problems, leading to a reduction on the short circuits and electrolyte leakages occurrence, and consequently to a significant increase in the battery cycle life [280]. In this sense, the addition of lithium nitride as an artificial SEI in the interface between the SPE and Li metal proved to inhibit the reactions between the electrolyte and the electrode. This layer avoids the use of lithium-ions for the formation of the common SEI, and it has the capacity to maintain its structure after several charge and discharge cycles, also avoiding the growth of lithium dendrites due to their excellent mechanical properties [259]. The use of UV copolymerization of polymers such as hexa(4-ethyl acrylate phenoxy) cyclotriphosphazene (HCP), (2-(3-(6-methyl-4-oxo-1,4-dihydropyrimidin-2-yl)ureido)ethyl methacrylate) (UPyMA), and poly(ethylene glycol) methyl ether methacrylate (PEGMA) is an effective way to combine the different properties of each polymer and create an SPE with high thermal and mechanical stability, self-healing, and fire retardant capacity, improving both the safety and the performance of LIBs [260]. Polymeric ionic liquids (PILs) can also be effectively applied for this purpose. The reversible ionic bonds created by the abundance of cations and anions existent in the polymer chains and in the interstitial ionic liquid enhance the self-healing capacity and creates a system of interconnected pathways for the lithium-ion migration, increasing the ionic conductivity of the LIB (Figure 10) [261]. Fast self-healing levels can be achieved by combining disulfide bonds with urea groups, particularly at high temperatures, without loss in battery performance, due to the effect of hydrogen and disulfide bonds, which return the material to its former state [262].

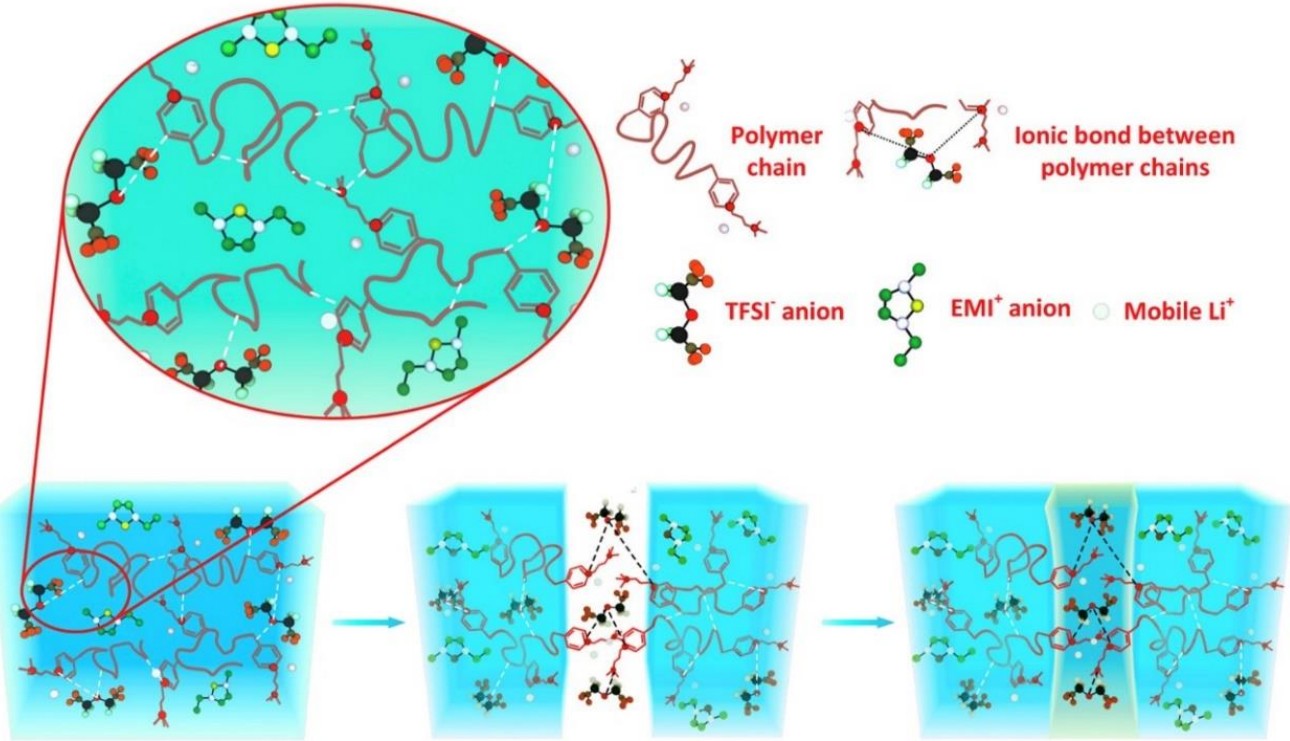

**Figure 10.** Self-healing mechanism in a PIL-based SPE [261].

Flame retardancy is also an important and needed development in the SPE technology, as it prevents the occurrence of incidents caused by short circuits, such as burning or explosion of the equipment. By doping the polymer matrix structure with phosphorous and silicon containing monomers, it is possible to increase the compatibility between the fillers and the matrix, which improves the overall stability even at high temperatures, enhancing the flame retardant performance [263].

More conventional approaches include the studies to increase battery performance, cycling stability, and to reduce the operation temperature of an SPE. In this context, the production of a PAES-g-PEG polymer blend, using room temperature ionic liquids as fillers lead to excellent cycling stabilities, particularly with the addition of nitrile functional groups to the matrix, with stability up to 500 cycles [264]. Ionic conduction can be increased when PVDF-HFP is added to the structure of an ETPTA UV cured polymer, with the formation of high ionic conduction pathways [266]. The use of MOFs promotes the intermolecular interactions in the polymer structure, which inhibits the crystallization of PEO, improving the ionic conductivity, particularly at high temperatures [267]. Ionic liquids are also an effective way to increase the ionic conductivity of an SPE, through the inhibition of the polymer crystallization [268,272]. The immobilization of lithium ionic liquids (LiIL) in a mesoporous silica structure creates a hybrid interface that provides fast $Li^+$ transportation kinetics without compromising the mechanical properties. The assembled batteries with this SPE showed good cycling performances at high temperatures [269]. Polymeric ionic liquids (PILs) can reduce the interfacial resistance in the battery, resulting in improved ionic conductivity, and consequently good cycling performance [270]. $BaTiO_3$ nanoparticles can be used to improve both the ionic conductivity and the dielectric constant of a PMMA based SPE, up to a maximum of 5166 [271]. The use of ultrasonic treatments in the PEO matrix proved to significant increase the ionic conductivity of the SPE by around 78% due to the breaking of PEO grains and reduction in the crystallinity [273]. Polymer-in-ceramic composites can also play a key role in the ionic conduction with fast $Li^+$ conduction and high lithium transference number [274]. The use of zeolites is still not much explored, but it shows promising results, particularly in the prolonged cycle life of the batteries [265]. The comparison between the two most used lithium salts (LiTFSI and LiClO₄) concluded that LiTFSI originates SPEs with the higher ionic conductivity, while the $Li^+$ transference number is increased by the addition of LiClO₄ [275].

Environmental issues are gaining significant relevance in modern society. Thus, the necessity to produce more sustainable devices has also reached the SPE field. In this context, the shift toward the use of environmental friendlier materials, both with the use of natural polymers as matrix, and by avoiding the use of organic solvents, is getting increasing interest [281]. The use of chitosan combined with a PEG plasticizer leads to an SPE with an excellent room temperature ionic conductivity, due to increasing polymer chain flexibility attributed to the plasticizer addition, making this a promising option for application in batteries [276]. Iota-Carrageenan [277], pectin, and guar gum [278] are other promising polymers that can achieve high room temperature ionic conductivity (in the order of $10^{-4}$ S·cm$^{-1}$). The use of water as solvent in a super hydrophilic PAA matrix doped with silica nanoparticles presents outstanding ionic conductivities above $10^{-2}$ S·cm$^{-1}$, being a promising candidate for aqueous rechargeable lithium-ion batteries [279].

## 5. Main Conclusions and Future Trends

Lithium-ion batteries (LIBs) are the most used energy storage system with increasing applicability on devices ranging from small sensors to large-scale and complex electric vehicles. The recent development in the materials used in the main three LIBs components, anode, cathode, and separator/electrolyte, have been presented and compared. These materials are focused on the resolution of the most frequent LIB issues, such as the ones related to their processability, safety, and stability, as well as to increase their performance. Furthermore, the environmental impact of materials and processes are gaining increasing relevance in this area. For the anode, the most studied active materials are carbon, metal

alloys, and silicon-based materials. Furthermore, conversion-type transition metals and their composite-based anode materials increased interest in recent years due to their high theoretical capabilities, low cost, and availability. Materials such as iron oxides and MOFs increase lithium storage capabilities and electrical conductivity, and they act as a buffer medium to reduce the volume change.

With respect to cathode active materials, the most used ones, including LFP, LCO, LMO, or LNMO, are being modified through doping with different elements, innovative synthesis methods have been developed, composites with different particles have been processed, particle size and morphology are being optimized, and performance is being improved by functionalization and coating. Efforts are also being made in the field of hybrid structures, using materials such as MXenes and MOFs, to improve the electrode's performance, with a focus on improving cycling behavior. Future trends in this area also include research in cobalt-free active materials, which will allow for the reduction of the battery costs, as cobalt is a scarce and costly component of batteries. These optimizations are focused to improve the electronic and thermal properties, to stabilize the particle with the electrolyte, and to improve the mechanochemical activation. Separator materials based on PP, PE, and PVDF, among other polymers, have been studied as microporous membranes, nonwoven membranes, and electrospun membranes. In addition to the structure, surface modification, composite membranes, and polymer blends have been studied showing improved lithium dendrite growth inhibition, improving thermal and safety properties, increasing the wettability, and improving interfacial issues. Studies of new environmentally friendly materials and SPE are increasing due to the commitment with advanced sustainable and safer materials in LIBs systems. Materials such as cellulose and silk, and fillers such as natural clay are some of these examples. The elimination of the liquid electrolyte in SPE strongly decreases the safety concerns visualized in the typical separators. Furthermore, SPE studies show that additional functionalities as battery shutdown, self-healing, and/or self-sensing ability can be implemented in those systems, strongly increasing battery characteristics, particularly at the safety level. In addition, initial studies demonstrate that natural polymers are a possible route for SPE development, once they allow for high room temperature ionic conductivity, without compromising sustainability. Thus, efforts to enhance the material properties of chitosan, iota-carrageenan, pectin, and guar gum have been made in order to be applied to LIBs.

The work on LIBs should always take into consideration improvements on all components of the batteries in order to achieve the best compatibility and improve the performance of the devices. Thus, the study of the different materials for a specific battery component must take into consideration the materials that will be integrated into the other components. The selection and combination of these materials will affect the overall performance of the system, aiming to solve the current performance, safety, stability, and environmental issues.

The development of LIB technology must also be accompanied with the advance in alternative energy storage systems, which allows for a higher diversity of options, limiting the over exploration of the same type of resources. These alternatives include sodium, potassium, and manganese-based batteries, which are areas of increasing research activity.

Thus, despite the strong success and implementation of Li-ion batteries in modern technology, efforts and the levels of materials must continue to provide a new generation of higher performance, safer, and environmentally friendlier batteries.

**Author Contributions:** Writing—original draft, Writing—review and editing, J.C.B., R.G., C.M.C. and S.L.-M.; Funding acquisition, S.L.-M. All authors have read and agreed to the published version of the manuscript.

**Funding:** Funding grants UID/FIS/04650/2020, UID/EEA/04436/2020 and UID/QUI/0686/2020; and project PTDC/FIS-MAC/28157/2017. Financial support grants SFRH/BD/140842/2018 (J.C.B.) and Investigator FCT Contracts CEECIND/00833/2017 (R.G.) and 2020.04028.CEECIND (C.M.C.) Financial support ELKARTEK and PIBA (PIBA-2018-06) programs.

**Institutional Review Board Statement:** Not applicable.

**Informed Consent Statement:** Not applicable.

**Acknowledgments:** The authors thank the FCT (Fundação para a Ciência e Tecnologia), POCH, European Union, Basque Government Industry and Education Departments for financial support.

**Conflicts of Interest:** The authors declare no conflict of interest.

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
