# Peer review of "Recent Advances on Materials for Lithium-Ion Batteries"

_energies, doi:10.3390/en14113145_

Round 1

Reviewer 1 Report

Overall, the review is well organised and the contents are very interesting. I got several comments which I think would be helpful for the improvement of presentation of the paper:

In the abstract and the introduction, the necessity of this review is missing. The authors should make it clear why this review is necessary, urgent, and important. For instance, are there any existing reviews on this topic? If so, what are the differences between them and this manuscript?

Fig. 1, the well-known name Goodenough was misspelled as Goodenoguh.

Some of the figures vary in front size and are of low quality (Fig. 3, Fig. 4, Fig. 5, Fig. 6, Fig. 8). Improvements are needed.

References should be given after the Figure caption.

line 68, XX represents what? 20th?

The literature study can be enriched, especially for the LIBs electrode materials relevant issues. e.g., volume expansion of electrode materials caused by diffusion-induced stresses during the normal charging and discharging of LIBs (e.g., static without external mechanical loading)  [1,2], and premature mechanical failure (deformation and fracture) of electrode materials caused by external mechanical loading exerted/imposed during the normal usage of electric vehicles (e.g, continuously running on rough road surfaces). In this respect, it would be helpful to refer to the following relevant papers:  [1] doi: 10.1142/S1758825113500403; [2] doi: 10.1149/1.2185287; [3] doi: 10.1016/j.electacta.2015.10.097

The most important part, discussion, is missing.

It is expected that more details (advantages, disadvantages, etc.) should be given for the comparison of the different materials and technologies, either incorporated in the related Table, or in the discussion.

Author Response

Overall, the review is well organised and the contents are very interesting. I got several comments which I think would be helpful for the improvement of presentation of the paper:

Thanks for the positive comments and suitable suggestions for improving the quality of the manuscript.

In the abstract and the introduction, the necessity of this review is missing. The authors should make it clear why this review is necessary, urgent, and important. For instance, are there any existing reviews on this topic? If so, what are the differences between them and this manuscript?

Yes, there exist different reviews in the area, but the novelty of this review is the focus on the recent advances of materials for battery components considering the strong growth of this field. The high number of published works in recent years makes necessary such a review and allows to compare this new information with other reviews that exist in the literature. New text in the abstract and the introduction sections has been added.

Fig. 1, the well-known name Goodenough was misspelled as Goodenoguh.

Thank you for the correction. We apologize for the mistake. The name has been corrected as suggested in figure 1.

Some of the figures vary in front size and are of low quality (Fig. 3, Fig. 4, Fig. 5, Fig. 6, Fig. 8). Improvements are needed.

The quality of the figures has been improved.

References should be given after the Figure caption.

It has been revised and changed in the manuscript.

line 68, XX represents what? 20th?

We do apologize for the mistake. It has been changed in the manuscript.

The literature study can be enriched, especially for the LIBs electrode materials relevant issues. e.g., volume expansion of electrode materials caused by diffusion-induced stresses during the normal charging and discharging of LIBs (e.g., static without external mechanical loading)  [1,2], and premature mechanical failure (deformation and fracture) of electrode materials caused by external mechanical loading exerted/imposed during the normal usage of electric vehicles (e.g, continuously running on rough road surfaces). In this respect, it would be helpful to refer to the following relevant papers:  [1] doi: 10.1142/S1758825113500403; [2] doi: 10.1149/1.2185287; [3] doi: 10.1016/j.electacta.2015.10.097

The most important part, discussion, is missing.

These issues have been included in section 3 in which new text has been added.

It is expected that more details (advantages, disadvantages, etc.) should be given for the comparison of the different materials and technologies, either incorporated in the related Table, or in the discussion.

More details on materials and technologies have been included in the manuscript.

Reviewer 2 Report

The development of lithium-ion batteries (LIBs) is one of the most important achievements over the past three decades. LIBs are the most commonly used ESS in modern society, mainly due to their high specific capacity, making them appropriate for small and light portable devices without limiting their performance. LIBs are also characterized by prolonged cycle life and no memory effects. These are important advantages that increased the use of LIBs, leading to a progressive replacement of previous technologies, such as nickel-cadmium and nickel-metal hydride batteries, which are less efficient, in particular for small device applications. In this paper, this review focuses on the different materials recently developed for the different battery components, anode, cathode and separator/electrolyte, in order to further improve LIB systems. Moreover, solid polymer electrolytes (SPE) for LIBs are also highlighted. Together with the study of new advanced materials, materials modification by doping or synthesis, combination of different materials, fillers addition, size manipulation or the use of high ionic conductor materials are also presented as effective methods to enhance the electrochemical properties of LIBs. Although the topic in this work was interesting, the presentation in this manuscript was very poor. This manuscript should be rejected for published in Energies. However, if the authors are willing to make the substantial revisions according to my comments, I would be glad to re-review this manuscript. Here are my detailed comments:

  1. The detailed literature review indicates efforts made by the authors. The coherence of the related work, however, is still not clear. It may help the authors by answering the following questions: Why are these works relevant? Which specific problems were addressed? How are the previous results related with the latest work? What are the outstanding, unresolved, research issues? Which of them has been solved by the proposed study? Answering the questions leads to the novelty of the proposed work naturally. Besides, the current one is nothing but a literature review. Why their work is important comparing to previous reports? I think this is essential to keep the interest of the reader.
  2. In Table 1, the authors should give the explanations for the difference of data collected from different sources.
  3. There is no Conclusions Part.
  4. Furthermore, SPE studies show that additional functionalities as battery shutdown or self-healing ability can be implemented in those systems, strongly increasing battery characteristics. To that, efforts to enhance materials properties for LIBs applicability of chitosan, iota-carrageenan, pectin and guar gum, have been made. Also, initial studies demonstrate that natural polymers are a possible route for SPE development, once allow for high room temperature ionic conductivity. The authors should give some explanation on above conclusions.
  5. Environmental issues related to energy consumption are mainly associated to the strong dependence on fossil fuels. To solve this issues, renewable energy sources systems have been developed, as well as advanced energy storage systems. Batteries are the main storage system related to mobility and they are applied in devices such as laptops, cell phones and electric vehicles. Energy shortage and environment pollution have seriously threatened people’s survival. Thus, the development of fuel cell has caught human attention. Besides lithium-ion batteries, proton exchange membrane fuel cells have attracted attention from energy devices such as portable, mobile and stationary devices, since it helps effective reductions of energy shortage and environment pollution, see [International Journal of Hydrogen Energy, 2018, 43(37):17880-17888; International Journal of Heat and Mass Transfer, 2019, 137:365-371]. Authors should introduce some related knowledge to readers. I think this is essential to keep the interest of the reader.
  6. Although the results look “making sense”, the authors should dig deeper in the results by presenting some in-depth discussion, such as implications of the results, such as possible application of them.
  7. Please, expand the conclusions in relation to the specific goals and the future work.
  8. English grammar and syntax has to be checked carefully throughout the manuscript. There are several grammatical mistakes in the manuscript and it is very difficult to follow anything if they are not corrected.

Author Response

The development of lithium-ion batteries (LIBs) is one of the most important achievements over the past three decades. LIBs are the most commonly used ESS in modern society, mainly due to their high specific capacity, making them appropriate for small and light portable devices without limiting their performance. LIBs are also characterized by prolonged cycle life and no memory effects. These are important advantages that increased the use of LIBs, leading to a progressive replacement of previous technologies, such as nickel-cadmium and nickel-metal hydride batteries, which are less efficient, in particular for small device applications. In this paper, this review focuses on the different materials recently developed for the different battery components, anode, cathode and separator/electrolyte, in order to further improve LIB systems. Moreover, solid polymer electrolytes (SPE) for LIBs are also highlighted. Together with the study of new advanced materials, materials modification by doping or synthesis, combination of different materials, fillers addition, size manipulation or the use of high ionic conductor materials are also presented as effective methods to enhance the electrochemical properties of LIBs. Although the topic in this work was interesting, the presentation in this manuscript was very poor. This manuscript should be rejected for published in Energies.

We thank the reviewer for the constructive comments and the suggestions for improvement.

However, if the authors are willing to make the substantial revisions according to my comments, I would be glad to re-review this manuscript. Here are my detailed comments:

  1. The detailed literature review indicates efforts made by the authors. The coherence of the related work, however, is still not clear. It may help the authors by answering the following questions: Why are these works relevant? Which specific problems were addressed? How are the previous results related with the latest work? What are the outstanding, unresolved, research issues? Which of them has been solved by the proposed study? Answering the questions leads to the novelty of the proposed work naturally. Besides, the current one is nothing but a literature review. Why their work is important comparing to previous reports? I think this is essential to keep the interest of the reader.

Thanks for the suggestions. Modifications have been implemented in the manuscript.

  1. In Table 1, the authors should give the explanations for the difference of data collected from different sources.

New text has been added in the manuscript.

  1. There is no Conclusions Part.

More details have been added in the manuscript.

  1. Furthermore, SPE studies show that additional functionalities as battery shutdown or self-healing ability can be implemented in those systems, strongly increasing battery characteristics. To that, efforts to enhance materials properties for LIBs applicability of chitosan, iota-carrageenan, pectin and guar gum, have been made. Also, initial studies demonstrate that natural polymers are a possible route for SPE development, once allow for high room temperature ionic conductivity. The authors should give some explanation on above conclusions.

The conclusion has been rearranged and more explanations have been included in the manuscript.

  1. Environmental issues related to energy consumption are mainly associated to the strong dependence on fossil fuels. To solve this issues, renewable energy sources systems have been developed, as well as advanced energy storage systems. Batteries are the main storage system related to mobility and they are applied in devices such as laptops, cell phones and electric vehicles. Energy shortage and environment pollution have seriously threatened people’s survival. Thus, the development of fuel cell has caught human attention. Besides lithium-ion batteries, proton exchange membrane fuel cells have attracted attention from energy devices such as portable, mobile and stationary devices, since it helps effective reductions of energy shortage and environment pollution, see [International Journal of Hydrogen Energy, 2018, 43(37):17880-17888; International Journal of Heat and Mass Transfer, 2019, 137:365-371]. Authors should introduce some related knowledge to readers. I think this is essential to keep the interest of the reader.

Thanks for the suggestion. New text and these references have been added in the manuscript

  1. Although the results look “making sense”, the authors should dig deeper in the results by presenting some in-depth discussion, such as implications of the results, such as possible application of them.

More explanations have been added.

  1. Please, expand the conclusions in relation to the specific goals and the future work.

The conclusion section has been rearranged and new text has been added.

  1. English grammar and syntax has to be checked carefully throughout the manuscript. There are several grammatical mistakes in the manuscript and it is very difficult to follow anything if they are not corrected.

The English language has been carefully revised.

Reviewer 3 Report

1. THERE ARE NUMEROUS TYPOS... e.g., is placed to avoid the physical contact between them, 88 avoiding short circuits.

Table 1 – Comparative analysis between different battery technologies [41, 42] [40, 136 43] [44]. - You can simple cite [40-44, 136]

2. General remark:

The manuscript is difficult to follow.

There are plenty review articles on Li-ion batteries, what is new here?  The title: "Recent advances on materials for lithium-ion batteries" would sugest that the authors will provide information about new approaches, but in practice, for example, in case of anode materials there is an introduction with 3-4 examples. Each description about anodes is constructed in the same way...

"Silicon based materials are widely used as anode active materials due to

Transition metals are widely used in anode electrodes for LIBs due to their...."

I would suggest to rewrite it.

3. Figure 2. Voltage range vs Li/Li+ as a function of the charge capacity of different anode active ma- 187
terials....looks like taken from other reviews. Where are references for that?

4. Figure 3, figure 4, figure 5 - low quality images.

5. There is only one reference under MDPI articles, which is also a self-citation.

6. Conclusions section:

metal-organic frameworks (MOF) - you should provide the abbreviations in previous sections where it appears for the first time.

7. There is an effort to design electrodes for Li-ion batteries based on hybrid structures which should be underlined in this review(MXenes, MOF, coatings, modifications and more...)

Author Response

  1. THERE ARE NUMEROUS TYPOS... e.g., is placed to avoid the physical contact between them, 88 avoiding short circuits.

This and other sentences have been rearranged in the manuscript

Table 1 – Comparative analysis between different battery technologies [41, 42] [40, 136 43] [44]. - You can simple cite [40-44, 136]

The citations have been corrected in the manuscript.

  1. General remark:

The manuscript is difficult to follow.

There are plenty review articles on Li-ion batteries, what is new here?  The title: "Recent advances on materials for lithium-ion batteries" would sugest that the authors will provide information about new approaches, but in practice, for example, in case of anode materials there is an introduction with 3-4 examples. Each description about anodes is constructed in the same way...

"Silicon based materials are widely used as anode active materials due to

Transition metals are widely used in anode electrodes for LIBs due to their...."

I would suggest to rewrite it.

We understand the comment.  These and other sentences have have been re-written in the manuscript as suggested.

  1. Figure 2. Voltage range vs Li/Li+ as a function of the charge capacity of different anode active ma- 187
    terials....looks like taken from other reviews. Where are references for that?

These references have been added in figure 2.

  1. Figure 3, figure 4, figure 5 - low quality images.

Thank you for the indication. The quality of these figures has been improved in the manuscript.

  1. There is only one reference under MDPI articles, which is also a self-citation.

More references to MDPI journals have been added in the manuscript.

  1. Conclusions section:

metal-organic frameworks (MOF) - you should provide the abbreviations in previous sections where it appears for the first time.

It has been changed in the manuscript.

  1. There is an effort to design electrodes for Li-ion batteries based on hybrid structures which should be underlined in this review (MXenes, MOF, coatings, modifications and more...)

More details in the conclusion section have been added in the manuscript.

Reviewer 4 Report

The author provided an overview of the anode, cathode and separator materials for Li-ion batteries. I recommend it for publication after the following issues being addressed.

  1. In Fig. 1, if the authors think the solid polymer electrolyte is a major advancement that is on par with the invention of Li-ion battery (which is arguable), at least, I would also include the solid ceramic electrolyte into the timeline.
  2. Line 85-87, the authors stated, "The current collectors are made of different metals and are placed in each electrode, leading to the potential difference, and allowing for the redox reactions to occur [20]." This is wrong, the potential difference of the battery comes from the cathode and anode, not from the current collectors. This has to be corrected.
  3. Line 173, the authors stated, "Anode active materials are crystalline structures of different types,...". This is inaccurate because the carbonaceous anode materials are non-crystalline. 
  4. Line 193-195, the authors stated, "One of the main disadvantages of graphite is the irreversible loss of capacity during the first charge-discharge cycle to form a stable interface between the electrolyte and the graphite, called the solid-electrolyte interface (SEI)". I would not say the formation of SEI is a disadvantage. In contrast, SEI prevents continuous degradation of electrolyte during cycling and enables long-term cycling of the battery. Without the SEI, a modern Li-ion battery will not work.
  5. There are many one-sentence paragraphs in this manuscript, e.g. lines 496-506. Please combine these into one paragraph. 
  6. In the main conclusions and future trends section, I don't see the future trends. Can the authors be more explicit about the future trend?    

Author Response

The author provided an overview of the anode, cathode and separator materials for Li-ion batteries.

Thank you for the positive comments on the manuscript and for the adequate suggestions for improve the quality of the manuscript.

I recommend it for publication after the following issues being addressed.

  1. In Fig. 1, if the authors think the solid polymer electrolyte is a major advancement that is on par with the invention of Li-ion battery (which is arguable), at least, I would also include the solid ceramic electrolyte into the timeline.

We understand the comment and the timeline has been changed and modified solid polymer electrolyte for solid-state battery to include the solid ceramic electrolyte. Thanks for the suggestion.

  1. Line 85-87, the authors stated, "The current collectors are made of different metals and are placed in each electrode, leading to the potential difference, and allowing for the redox reactions to occur [20]." This is wrong, the potential difference of the battery comes from the cathode and anode, not from the current collectors. This has to be corrected.

We do agree, of course. Sorry for any possible misunderstanding. We have changed the sentence for clarity.

  1. Line 173, the authors stated, "Anode active materials are crystalline structures of different types,...". This is inaccurate because the carbonaceous anode materials are non-crystalline. 

This word has been eliminated.

  1. Line 193-195, the authors stated, "One of the main disadvantages of graphite is the irreversible loss of capacity during the first charge-discharge cycle to form a stable interface between the electrolyte and the graphite, called the solid-electrolyte interface (SEI)". I would not say the formation of SEI is a disadvantage. In contrast, SEI prevents continuous degradation of electrolyte during cycling and enables long-term cycling of the battery. Without the SEI, a modern Li-ion battery will not work.

This sentence has been changed in the manuscript.

  1. There are many one-sentence paragraphs in this manuscript, e.g. lines 496-506. Please combine these into one paragraph. 

These paragraphs have been combined as suggested.

  1. In the main conclusions and future trends section, I don't see the future trends. Can the authors be more explicit about the future trend?    

The future trends have been included in the manuscript.

Reviewer 5 Report

This review summarized different components including anode, cathode and separator/electrolyte recently developed for the LIB battery. The content of this review is complete and the relevant works are comprehensively described. However, some issues still remain to be addressed as listed below:

  1. In the line 69-70, the author stated that the safety problem of lithium dendrite was solved by the development of cathode. It should be the development of carbon-based anode material that avoid the use of lithium metal anode. The author need to check the development of LIB history to confirm it.
  2. The equation 1 in Line 105 should be two-way.
  3. The references cited in Table 1 is confusing. It is better to add a new column to cite the relevant references.
  4. The abbreviation in the review should be checked again. Some abbreviation appeas two times, and the following content should use the corresponding abbreviation if it has been abbreviated in the previous part as in Table 3.
  5. In the 3.1.3 part, some materials including Si and Sn based materials are not the conversion type. The author need to check and confirm it again.
  6. In the cathode materials part, there is little summary about low cobalt or cobalt-free cathode, which is now a reasearch focus. It is better to add relevant discussion about it.
  7. The form of references are not consistent (for example, the title of the reference), and need to be adjusted.
  8. The superscript and subscript througout the review need to be checked again.

Author Response

This review summarized different components including anode, cathode and separator/electrolyte recently developed for the LIB battery. The content of this review is complete and the relevant works are comprehensively described.

Thanks to the reviewer for the positive comments.

However, some issues still remain to be addressed as listed below:

  1. In the line 69-70, the author stated that the safety problem of lithium dendrite was solved by the development of cathode. It should be the development of carbon-based anode material that avoid the use of lithium metal anode. The author need to check the development of LIB history to confirm it.

The reviewer is right and the phrase has been changed in the manuscript.

  1. The equation 1 in Line 105 should be two-way.

Equation 1 has been corrected.

  1. The references cited in Table 1 is confusing. It is better to add a new column to cite the relevant references.

We understand and the references have been changed in the manuscript. Some information regarding the same technology was found in different references, making it confusing when placed in in the same column.

  1. The abbreviation in the review should be checked again. Some abbreviation appeas two times, and the following content should use the corresponding abbreviation if it has been abbreviated in the previous part as in Table 3.

The abbreviations have been revised.

  1. In the 3.1.3 part, some materials including Si and Sn based materials are not the conversion type. The author need to check and confirm it again.

It has been checked and confirmed.

  1. In the cathode materials part, there is little summary about low cobalt or cobalt-free cathode, which is now a reasearch focus. It is better to add relevant discussion about it.

We do agree that this issue is very important. New text about this issue has been included in the manuscript.

  1. The form of references are not consistent (for example, the title of the reference), and need to be adjusted.

All references have been revised and modified.

  1. The superscript and subscript througout the review need to be checked again.

These errors types have been revised and corrected.

Round 2

Reviewer 1 Report

Thank you for the response, I have no further comments.

Author Response

Thank you for the positive feedback.

Reviewer 2 Report

Very good. It is ok.

Author Response

Thank you for your comment.

Reviewer 3 Report

Figure 1  - line 48&49 are almost the same

Table 1 - I think it will be accurately to point references to each type of batteries

Sections Funding and Acknowledgments are almost the same.

Author Response

Figure 1  - line 48&49 are almost the same

We appreciate the reviewer observation. It was a mistake with the track changes functionality of word.  It has been corrected in the manuscript.

Table 1 - I think it will be accurately to point references to each type of batteries

A column for the references was added to Table 1.

Sections Funding and Acknowledgments are almost the same.

These sections were improved.

Reviewer 4 Report

My previous comments have been addressed. I recommend it for publication. 

Author Response

Thank you for the positive feedback.

Reviewer 5 Report

The authors have made great revision to the manuscript based on the reviewers' s advice.

Author Response

We thank the reviewer for the positive comment.